# FROM SEEING TO DOING: BRIDGING REASONING AND DECISION FOR ROBOTIC MANIPULATION

**Yifu Yuan**[1], **Haiqin Cui**[1,†], **Yibin Chen**[1,†], **Zibin Dong**[1], **Fei Ni**[1], **Longxin Kou**[1]
**Jinyi Liu**[1], **Pengyi Li**[1], **Yan Zheng**[1], **Jianye Hao**[1,*]

[1]College of Intelligence and Computing, Tianjin University
[†]HC and YC are co-second authors.
[*]Corresponding authors: Jianye Hao (jianye.hao@tju.edu.cn).

## ABSTRACT

Achieving generalization in robotic manipulation remains a critical challenge, particularly for unseen scenarios and novel tasks. Current Vision-Language-Action (VLA) models, while building on top of general Vision-Language Models (VLMs), still fall short of achieving robust zero-shot performance due to the scarcity and heterogeneity prevalent in embodied datasets. To address these limitations, we propose **FSD** (*From Seeing to Doing*), a novel vision-language model that generates intermediate representations through spatial relationship reasoning, providing fine-grained guidance for robotic manipulation. Our approach combines a hierarchical data construction pipeline for training with a self-consistency mechanism that aligns spatial coordinates with visual signals. Through extensive experiments, we comprehensively validated FSD's capabilities in both "*seeing*" and "*doing*", achieving outstanding performance across 8 benchmarks for general spatial reasoning and embodied reference abilities, as well as on our proposed more challenging benchmark **VABench**. We also verified zero-shot capabilities in robot manipulation, demonstrating significant performance improvements over baseline methods in both SimplerEnv and real robot settings. Experimental results show that FSD achieves 40.6% success rate in SimplerEnv and 72% success rate across 8 real-world tasks, outperforming the strongest baseline by 30%. More visualizations and datasets are available on website.

## 1 INTRODUCTION

A driving force behind robotics research is the pursuit of generalization: creating agents capable of versatile action across diverse robotic platforms, extending beyond familiar tasks, objects, and environments while adapting to dynamic visual inputs. Current approaches (Kim et al., 2024; Brohan et al., 2023; Ni et al., 2025) leverage pre-trained Vision-Language Models (VLMs) and transform them into Vision-Language-Action Models (VLAs) using large-scale embodied datasets. The intention is to capitalize on the generalization capabilities of VLMs pre-trained on internet-scale data, with the hope that resulting VLAs will adapt to novel scenarios. However, empirical evidence (Zheng et al., 2024; Zawalski et al., 2024; Liu et al., 2024e) demonstrates that this approach falls short of achieving strong zero-shot performance on completely novel tasks.

We attribute the limited generalization in existing VLA-based systems to two fundamental challenges: data *scarcity* and *heterogeneity*. Firstly, robotics data remains limited compared to the vast scale of language and vision datasets, preventing the emergence of similar scaling laws (Kaplan et al., 2020; Lin et al., 2024a). Secondly, the heterogeneity of robotic embodiments (Wang et al., 2024) introduces significant variations in action spaces and physical interactions, making the end-to-end supervised learning of a direct vision-to-action mapping a potentially unrealistic path toward true generalization. To bridge this generalization gap, the community has explored several paradigms. **End-to-end VLAs** (Black et al., 2024; Brohan et al., 2023) attempt a direct mapping from multimodal inputs to low-level actions, but the disconnect between pre-trained cyberspace data and physical action modalities can lead to knowledge forgetting and task conflicts. **Modular Methods** (Huang et al., 2024; Liu et al., 2024a) chain together specialized models for tasks like object detection and grasping,

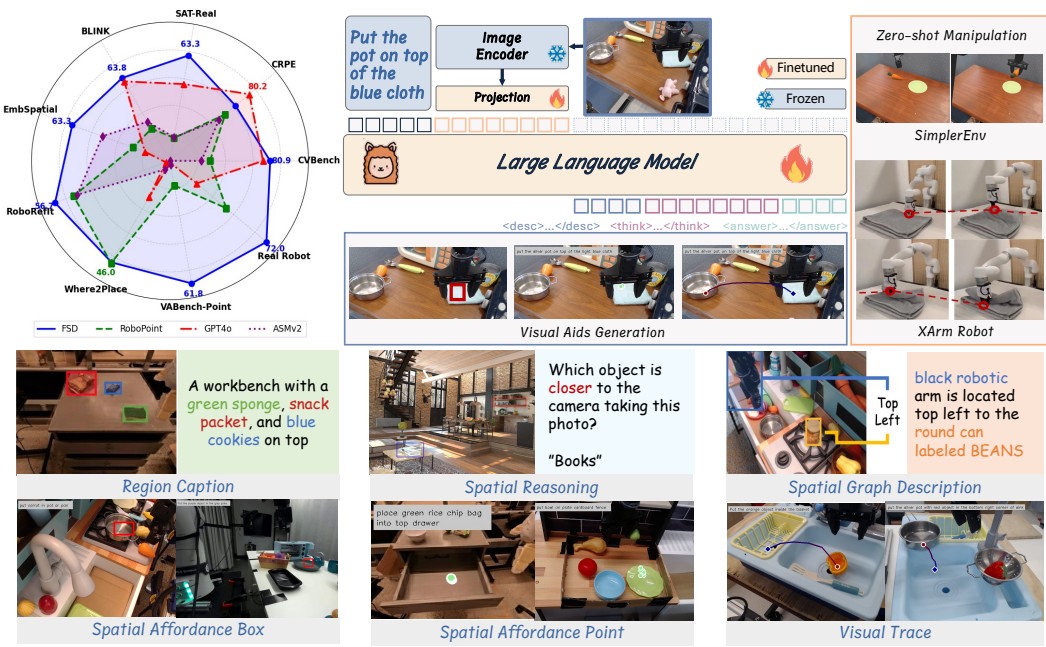

**Figure 1: Overview of FSD**. FSD unlocks visual aids reasoning and generation through Spatial Relationship-Focused CoT, demonstrating exceptional generalization capabilities that enable zero-shot robot manipulation and achieving remarkable performance across multiple benchmarks.

but these pipelines are prone to cascading failures, are difficult to tune, and suffer from high inference latency and a lack of holistic scene understanding. **Affordance Methods** (Yuan et al., 2024b; Xu et al., 2025; Zhou et al., 2025a;b) offer a promising direction by predicting intermediate visual aids (e.g., grasp points). However, they remain limited, often providing aids that are not comprehensive enough for complex decision-making and predicting raw coordinates without an explicit reasoning process, which hinders their ability to anchor instructions to the correct semantic entities.

We argue that the key to generalization lies not merely in predicting visual aids, but in first conducting explicit reasoning over the spatial and semantic context to generate an expressive, embodiment-independent intermediate representation. This representation, composed of visual aids like spatial affordance boxes and visual traces, provides compact yet powerful information for downstream decision-making. To this end, we propose **FSD (*From Seeing to Doing*)**, a novel framework that generates these visual intermediate representations through structured spatial reasoning (Fig. 1). FSD comprises three key components: ❶ Spatial relationship-focused visual chain-of-thought (**SrCoT**): This core mechanism treats visual aid generation as a reasoning process. It first performs a step-by-step textual analysis of the scene to identify relevant objects and infer their spatial relationships. Only after this semantic and spatial understanding is established does it generate a comprehensive set of object-centric visual aids. ❷ Hierarchical data construction pipeline: We integrate large-scale embodied datasets with commonsense data to build the foundational capabilities required for embodied reasoning, enabling a weak-to-strong capability enhancement during training. ❸ Self-consistency mechanism: This aligns the model's understanding and generation by binding spatial coordinates with specific visual signals, ensuring the generated aids are well-grounded. Besides, we manually annotated 300 images from diverse real-world scenarios to create the realistic and complex Visual Aids Generation Benchmark (VABench) to evaluate our approach.

FSD is an enhanced affordance-based VLA that generalizes effectively to new instructions and scenes through its reasoning abilities. Our contributions include: 1) A novel paradigm where VLM reasoning generates versatile visual aids, enabling either direct open-loop control or serving as the high-level planner in a hierarchical closed-loop policy; 2) The SrCoT method with a self-consistency mechanism for multi-step reasoning and guiding zero-shot manipulation; 3) Our weak-to-strong spatial reasoning and visual aids datasets, along with VABench; 4) Superior performance across 8 benchmarks in spatial reasoning, free space reference, and visual aids generation, achieving 40.6% zero-shot success in SimplerEnv and 72% in 8 real-world tasks, outperforming the RoboPoint baseline by 30%.

## 2 RELATED WORK

**Spatial Understanding and Reasoning with VLMs** Spatial understanding and reasoning, the ability to infer spatial relationships beyond 2D images, is crucial for embodied AI applications like navigation (Song et al., 2024; Hong et al., 2023) and manipulation (Yuan et al., 2024b). Recent advances such as SpatialVLM (Chen et al., 2024), SpatialRGPT (Cheng et al., 2024), and SpatialBot (Cai et al., 2024) have enhanced geometric reasoning by explicitly modeling spatial primitives and relationships. FSD specifically targets embodied manipulation by introducing a novel SrCoT mechanism and a sophisticated self-consistency alignment technique to advance spatial reasoning for complex tasks.

**Visual Chain-of-Thought Reasoning** Chain-of-thought reasoning (Wei et al., 2022) has significantly enhanced LLM performance by structuring step-by-step thought processes. This paradigm has been extended to multimodal domains to tackle complex visual reasoning challenges (Mitra et al., 2024; Zheng et al., 2023a). Approaches like Shikra (Chen et al., 2023) and VoCoT (Li et al., 2024d) anchor reasoning in specific visual regions, while EmbodiedCoT (Zawalski et al., 2024) pioneered its application in embodied AI by fine-tuning VLAs to generate intermediate reasoning steps. In contrast, FSD advances this area by uniquely employing spatial relationship graphs as reasoning anchors, creating a more structured and grounded visual-spatial reasoning process.

**Visual Aids Empowered Robotic Manipulation** Extracting embodiment-agnostic visual aids to enhance training efficiency has emerged as a promising paradigm in robotic manipulation. Numerous studies (Bharadhwaj et al., 2024; Wen et al., 2023; Xu et al., 2024; Zheng et al., 2024; Yuan et al., 2024a; Ni et al., 2024) have explored robotic policy learning based on visual traces, though most remain confined to specific tasks with cross-embodiment applicability. LLaRVA (Niu et al., 2024) advances this field by predicting visual traces to better align visual and action spaces for robot learning, compiling an impressive large visual trace dataset, yet struggles to generalize to novel downstream tasks without task-specific fine-tuning. Spatial affordance represents another effective visual aid, with several works (Yuan et al., 2024b; Mo et al., 2021; Qin et al., 2020; Song et al., 2024; Ji et al., 2025; Yang et al., 2025; Li et al., 2025; Yuan et al., 2026) demonstrating its utility in robotic manipulation tasks. Consequently, FSD adopts a reasoning-driven paradigm, leveraging the VLM's general world knowledge for visual aids generation, thereby enhancing zero-shot generalization to unseen scenarios.

## 3 BRIDGING REASONING AND DECISION VIA VISUAL AIDS GENERATION

To harness VLM perception for cross-domain generalization, we propose **FSD** (*From Seeing to Doing*). Our approach generates embodiment-agnostic visual aids (spatial affordances and traces) via a novel reasoning mechanism: the Spatial Relationship-Focused CoT (SrCoT). SrCoT uniquely leverages object coordinates and their spatial relationships as reasoning anchors to produce precise visual aids. Then, FSD are trained with a progressive weak-to-strong data construction pipeline, further refined by a self-consistency alignment mechanism.

### 3.1 DEFINITION OF VISUAL AIDS

As shown in Fig. 2, FSD utilizes three visual aids of increasing complexity, all defined within normalized image coordinates $\mathbf{x} = (p, q) \subset \mathbb{R}^2$: **Spatial affordance boxes** $\mathcal{B} = [x_1, y_1, x_2, y_2]$ define target placement regions, including unmarked free space beyond standard detector capabilities; **Spatial affordance points** $\mathcal{P} = \{(x_i, y_i) \mid i = 1, 2, ..., n\}$ provide more precise and flexible targets; and **Object-centric visual traces** $\boldsymbol{\tau} = \{\mathbf{x}_t \mid t = 1, 2, ..., T\}$, which are ordered coordinate sequences describing complex, cross-embodiment manipulation trajectories. Our choice of an object-centric, rather than agent-centric, approach is crucial: it circum-

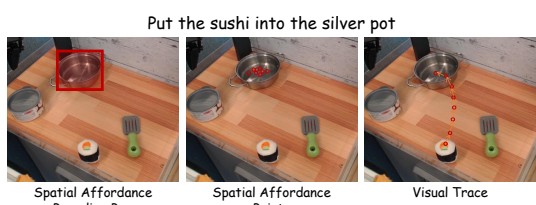

Figure 2: Definition of visual aids with increasing complexity. FSD leverages spatial affordance boxes for region-level guidance, points for coordinate-level precision, and visual traces for temporal sequences in an object-centric coordinate system.

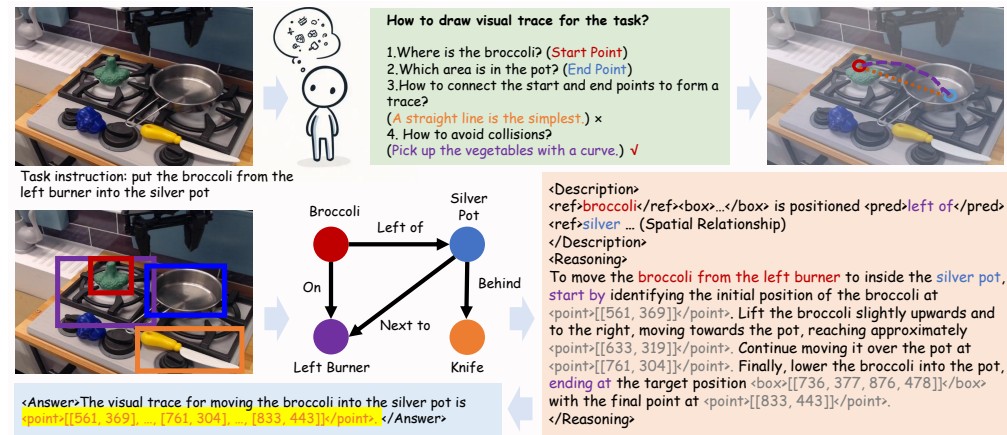

**Figure 3:** Inspired by the process of human reasoning, FSD uses a spatial relationship graph as an anchor to derive a visual chain-of-thought reasoning process for visual trace generation.

vents the limitations of heterogeneous embodied data and leverages general visual datasets to achieve robust generalization.

## 3.2 SPATIAL RELATIONSHIP-FOCUSED VISUAL CHAIN-OF-THOUGHT

To enable VLMs to generate spatial visual aids, a direct approach is supervised fine-tuning (Niu et al., 2024; Yuan et al., 2024b; Li et al., 2024b; Zawalski et al., 2024) using these aids as a new action space or employing generative models (Shridhar et al., 2024; Xu et al., 2024). However, the heterogeneity and scarcity of embodied datasets limit this method. Directly aligning RGB images with coordinate points is challenging and prone to overfitting, hindering generalization. *How can we stimulate VLMs' spatial reasoning abilities to guide the generation rather than merely relying on extensive demonstrations?* Inspired by human cognition (Fig. 3 (Top)), when executing tasks like "putting broccoli into a pot," humans first locate relevant objects, then plan movement paths based on relative positions while considering feasibility and obstacles. During this process, humans construct reasoning chains, repeatedly referencing object positions and establishing spatial relationships.

Based on these considerations, we introduce Spatial Relationship-Focused Visual Chain-of-thought (SrCoT). This approach guides VLMs to generate visual aids through structured reasoning based on spatial relationship graphs. SrCoT consists of two essential phases: ❶ **Description**: We generate object-centric region captions establishing a task-relevant spatial relationship graph where nodes represent objects with their coordinates and edges denote relative relationships (above, below, left, right, etc.). ❷ **Reasoning**: Using the spatial relationship graph as anchor points, we determine start and end coordinates through object references and free space reasoning, then iteratively derive intermediate points with explicit logical connections between steps. Thus, we prescribe a templated reasoning path for VLMs, enabling FSD to perform analogical reasoning in the spatial domain. While VLMs struggle to directly map future actions to image coordinates, our method leverages known object relationships as reference points for multi-hop analysis, simplifying the reasoning process. Fig. 3 (Bottom) demonstrates a complete reasoning sequence. This step-by-step SrCoT approach, though powerful, fundamentally depends on precise spatial understanding capabilities. To improve the stability and reliability of reasoning paths and reduce model hallucinations, SrCoT requires the model to generate coordinates in a specified format and bind them to objects while performing object-centric reasoning (Wang et al., 2025; Li et al., 2024d). We use *<ref>* to mark objects, and *<point>* and *<box>* to mark points and boxes, respectively, ensuring that each object is strictly bound to its corresponding coordinates. This explicit visual-spatial coordinate alignment enhances the FSD 's understanding of the spatial positions and relationships of objects. All coordinates are treated as text and are discretized as integers normalized between 0 and 999.

## 3.3 WEAK-TO-STRONG CAPABILITY DATASET CONSTRUCTION PIPELINE

The SrCoT mechanism places significant demands on VLMs, requiring capabilities like precise grounding, spatial understanding, and complex instruction following where mainstream models (Liu

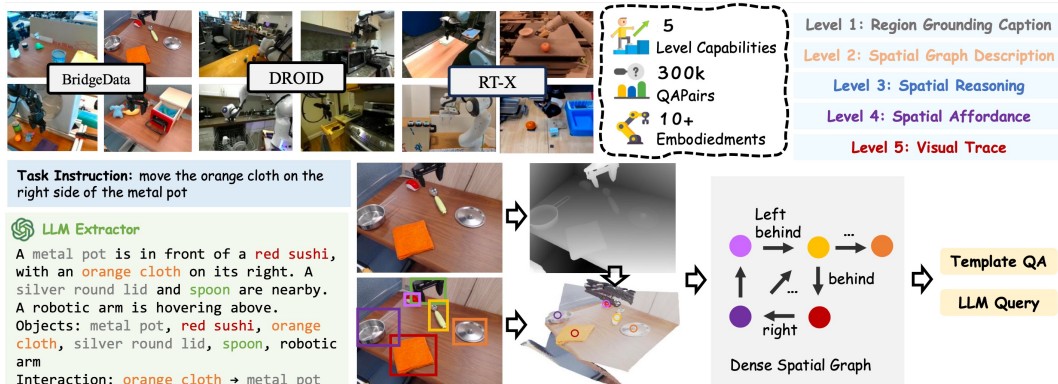

**Figure 4:** FSD screens data from large-scale embodied datasets, generates ground truth spatial relationship graph. We finally collected 300K data for 10+ embodiments with 5-level capabilities.

et al., 2024c; Lin et al., 2024b) show limitations. To address this, we developed a progressive, weak-to-strong data pipeline to cultivate these abilities hierarchically. FSD encompasses five hierarchical capability levels: ❶ **Region Grounding** enables robots to focus on key objects in scenes. Although grounding capabilities have been broadly integrated into current VLMs (Chen et al., 2023; You et al., 2023), understanding various small objects and complex scenes for embodied tasks is still limited; ❷ **Spatial Relationship** understanding establishes prerequisite knowledge for spatial reasoning, forming the anchor points for SrCoT; ❸ **Spatial Reasoning** builds upon these foundations to perform multi-hop analysis of object positions and relationships; and finally, ❹ **Spatial Affordance Generation** and ❺ **Visual Trace Generation** create actionable spatial guidance. Notably, SrCoT functions as a general visual-spatial reasoning mechanism applicable beyond visual traces to diverse spatial reasoning tasks. Through hierarchical spatial capability training, we enhance VLMs' general spatial reasoning abilities, extending well beyond just embodied domains.

**Automatic Dataset Construction:** We constructed a 300K Supervised Fine-Tuning (SFT) dataset by processing demonstrations from large-scale sources like BridgeDataV2 (Walke et al., 2023), RT-X (O'Neill et al., 2023), and Droid (Khazatsky et al., 2024). It first produces grounding data (**Level ❶**) by using a VLM to nominate objects and a vision model to extract their bounding boxes. Next, to establish spatial relationships (**Level ❷**), we reconstruct 3D scene graphs using Metric3Dv2 (Hu et al., 2024) and WildCamera (Zhu et al., 2024) to infer the relative positions of objects. While precise depth accuracy was not required for inferring positional relationships, to ensure high-quality data, we only processed object pairs exhibiting a relative depth gap of at least 20%. Then, these 3D scene graphs serve as a basis for a VLM to automatically generate complex spatial reasoning Q&A pairs (**Level ❸**). A core aspect of the FSD dataset is the visual aids generation. We employ a simple method with successful human demonstrations from embodied datasets and infer the process from the results. Spatial affordance represents the designated completion area for manipulation tasks. To create spatial affordance labels (**Level ❹** Dataset), we extract the manipulated object's final position from the terminal frame, combine it with reference object positioning, calculate the precise affordance region, and re-render this information onto the initial frame. For visual trace generation (**Level ❺** Dataset), we employ a two-stage approach: first applying self-supervised keypoint extraction (Huang et al., 2024) to identify grasp points on manipulated objects, then utilizing Cotracker (Karaev et al., 2024) to capture temporal dynamics from human demonstrations, subsequently projecting these trajectories onto the initial frame. Throughout this process, we employed strict rule-based filters and continually validated our approach against manually annotated test sets, iteratively refining our filtering criteria based on empirical feedback to ensure the resulting dataset met our quality requirements. The dataset presentation, data filtering process, and prompts used to generate the data are provided in App.A.

## 3.4 SELF-CONSISTENT ALIGNMENT FOR SPATIAL UNDERSTANDING AND GENERATION

High-quality SFT datasets enable VLMs to generate visual aids, yet these models struggle to understand the physical meaning of such annotations since coordinate spaces never appeared in pretraining data. The alignment between image coordinates and actual spatial positions presents a significant challenge. Therefore, we propose a self-consistency mechanism to further align FSD capabilities in

spatial understanding and generation. We frame generation tasks inversely as understanding problems: if the forward task requires inferring visual trace $\boldsymbol{\tau}$ from an image $X_v$ and task instruction $X_q$, i.e. $(X_v, X_q) \to \boldsymbol{\tau}$, we construct the inverse task of predicting possible instructions given an image and visual traces $(X_v, \boldsymbol{\tau}) \to X_q$. This bidirectional approach helps the model comprehend spatial coordinates' meanings and aligns coordinate space with image-text modalities, unifying visual aids as both understanding and generation signals while enhancing FSD spatial reasoning capabilities.

## 4    TRAINING AND ACTION EXECUTION OF FSD

**Training:** We adopt the instruction tuning pipeline from LLaVA-1.5 (Liu et al., 2024c), as illustrated in Fig. 1. FSD's architecture features a frozen CLIP-ViT-L (Gao et al., 2024) image encoder and a Vicuna-13B (Zheng et al., 2023b) LLM, which are connected by a trainable linear projector. This projector maps visual tokens into the language embedding space. We initialize our model from ASMv2 (Wang et al., 2025), which provides a strong foundation with its inherent grounding and relation conversation abilities. The training process unfolds in two stages: **General Spatial Reasoning Enhancement**: In the first stage, we use our Level 1-3 data to cultivate the model's core embodied spatial reasoning. To prevent catastrophic forgetting and retain pre-trained knowledge, we employ a mixed-data strategy, combining our robotics data with a diverse 1.4M sample of general visual question answering (VQA) and internet data. This joint training is crucial for developing robust embodied capabilities. **Visual Aids Generation and Understanding**: Using data from levels 4-5 with the self-consistency mechanism, we specifically train visual aids generation and understanding abilities. FSD predicts a fixed set of 8 points for simplification when generating spatial visual traces. Additional training details and the summary of mixture datasets are provided in App.B.

**Action Execution:** FSD can reason from initial or intermediate task steps, freely selecting needed visual aids. When using bounding boxes, we sample the center as the target point; with affordance points, we directly sample one point. For visual trace execution, we first generate 2D visual traces $\boldsymbol{\tau}$ and obtain preliminary depth information from depth cameras. Following the pinhole camera model, we employ depth-based back-projection to map these to 3D space, yielding $\boldsymbol{\tau}^{3d} = \{\mathbf{x}_t^{3d} \mid t = 1, 2, ..., T\}$. Next, based on the spatial position of the first point $\mathbf{x}_1$, we query GraspNet's (Fang et al., 2020) grasp candidates $G$ to match the nearest grasp pose $G^*$. For relatively fixed scenes, we may also use predetermined grasp poses. Subsequently, we optimize the path trajectory using gradient descent-based interpolation, generating complete motion trajectories in SE(3) space, enabling the robotic arm to follow the 3D visual trajectory. When using only spatial affordance, we utilize CuRobo (Sundaralingam et al., 2023) as the motion planner to determine execution trajectories $T$ based on the target position $\mathcal{P}$. More details are provided in App.C. Unlike methods such as LLARVA (Niu et al., 2024) and EmbodiedCOT (Zawalski et al., 2024) which also utilize visual auxiliary aids, FSD transforms prediction tasks into reasoning tasks, better leveraging visual-spatial common knowledge without requiring scenario-specific fine-tuning.

## 5    VISUAL AIDS GENERATION BENCHMARK

Existing datasets for evaluating visual aid generation are scarce, with benchmarks like Where2Place (Yuan et al., 2024b) being limited to simple instructions and a notable absence of benchmarks for visual trace prediction. To address these gaps, we introduce the Visual Aids Generation Benchmark (**VABench**), comprising 300 problems manually annotated from diverse real-world datasets (OXE (O'Neill et al., 2023), BridgeData (Walke et al., 2023), and Droid (Khazatsky et al., 2024)). VABench requires models to infer visual aids from natural language instructions that mimic everyday commands and is evaluated across two main tasks. For spatial affordance (*VABench-P*), we measure the proportion of points falling within target regions, sampling uniformly from predicted bounding boxes when necessary. For visual traces (*VABench-V*), we compute the Mean Absolute Error (MAE) and Root Mean Square Error (RMSE) against the ground truth, normalizing coordinates to a 1000×1000 space and using interpolation to ensure consistent evaluation. Recognizing that many instructions admit multiple valid solutions, we supplement these quantitative metrics with a qualitative LLM Score (1-10) assigned by an MLLM based on detailed assessment criteria that simulate human judgment. A complete description of our evaluation procedures is provided in App.D.

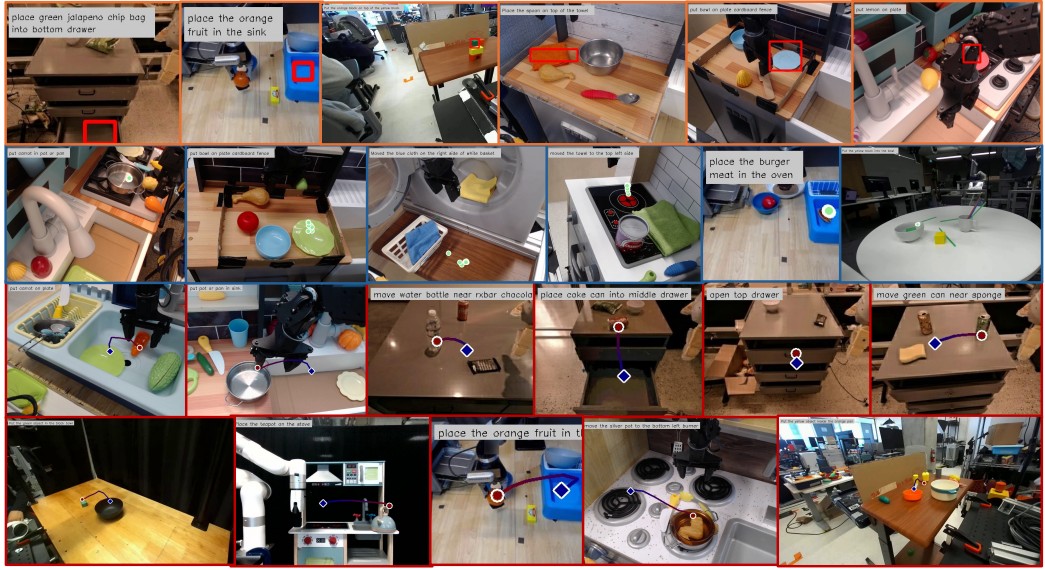

**Figure 5:** FSD directly generates visual aids based on task instructions for novel tasks and scenarios. *1st* row: affordance bounding boxes; *2nd* row: affordance points; *3rd* and *4th* rows: visual traces.

**Table 1:** Performance comparison on 5 spatial reasoning benchmarks. **Bold** and underlined values show best and second-best performance among open-source models.

| | CVBench | | | | | CRPE | | | | | SAT | | BLINK | | | | | EmbSp | Rank |
|---|---|---|---|---|---|---|---|---|---|---|---|---|---|---|---|---|---|---|---|
| | Count | 2DRel | 3DDep | 3DDis | Avg. | Exist. | Subj. | Pred. | Obj. | Avg. | Val | Real | Count | MV | RelDepth | SpRel | Avg. | Test | |
| *Closed-source models* | | | | | | | | | | | | | | | | | | | |
| GPT-4V | 62.4 | 71.1 | 79.8 | 68.3 | 70.4 | 90.6 | 76.7 | 65.1 | 68.5 | 75.2 | 44.8 | 50.7 | 60.8 | 55.6 | 59.7 | 72.7 | 62.2 | 36.1 | - |
| GPT-4o | 65.9 | 85.5 | 87.8 | 78.2 | 79.4 | 93.3 | 81.9 | 71.8 | 73.6 | 80.2 | 49.4 | 57.5 | 49.2 | 60.2 | 74.2 | 69.2 | 63.2 | 49.1 | - |
| *Open-source models* | | | | | | | | | | | | | | | | | | | |
| LLaVA-1.5-13B | 58.2 | 46.6 | 53.0 | 47.8 | 51.4 | 88.7 | 57.4 | 54.2 | 55.2 | 63.9 | 51.4 | 41.6 | 45.0 | 41.4 | 53.2 | 69.9 | 52.4 | 35.1 | 4.8 |
| SAT-Dynamic-13B | 61.5 | **89.7** | 80.7 | 73.0 | 76.2 | 87.5 | 60.6 | 57.6 | 65.2 | 67.7 | **87.7** | 54.9 | 35.8 | 44.4 | **73.4** | 66.4 | 55.0 | 51.3 | 2.8 |
| RoboPoint-13B | 56.5 | 77.2 | 81.5 | 57.7 | 68.2 | 93.2 | 66.3 | 62.4 | **70.9** | 73.2 | 53.3 | 46.6 | 48.3 | 44.4 | 62.1 | 65.7 | 55.1 | 51.4 | 2.8 |
| ASMv2-13B | 58.9 | 68.9 | 68.9 | 68.9 | 66.4 | 92.1 | 69.2 | 59.0 | 65.3 | 71.4 | 63.9 | 46.7 | 59.2 | 44.4 | 56.5 | 65.0 | 56.3 | 57.4 | 3.1 |
| FSD-13B | **62.4** | 86.5 | **88.0** | **86.7** | **80.9** | **94.0** | **75.2** | **65.1** | 70.4 | **76.2** | 73.2 | **63.3** | **60.0** | **46.6** | 70.2 | **78.3** | **63.8** | **63.3** | **1.3** |

# 6 EXPERIMENTS

We evaluated FSD across two dimensions: ***Seeing*** and ***Doing***. For *Seeing*, we tested its general spatial reasoning and visual aids generation capabilities. For *Doing*, we conducted zero-shot manipulation experiments in both SimplerEnv (Li et al., 2024c) simulation and real-world xArm robotic platforms to assess its practical generalization performance.

## 6.1 EVALUATION OF SPATIAL UNDERSTANDING AND REASONING CAPABILITIES

**Benchmarks and Baselines.** *General Spatial Reasoning Capabilities:* We evaluated general spatial reasoning capabilities using five popular benchmarks: CVBench (Tong et al., 2024), BLINK (Fu et al., 2024), CRPE (Wang et al., 2025), SAT (Ray et al., 2024), and EmbSpatial-Bench (Du et al., 2024). These benchmarks encompass 15 subtasks measuring various spatial competencies. We included two leading closed-source models: GPT-4o and GPT-4V as performance reference. Subsequently, we conducted comparative analyses against other open-source spatial enhanced MLLMs, including LLaVA-1.5 (Liu et al., 2023a), SAT-Dynamic (Ray et al., 2024), RoboPoint (Yuan et al., 2024b), and ASMv2 (Wang et al., 2025), all with 13B parameters. *Object and Free Region Reference Capabilities:* We assessed embodied spatial capabilities using the RoboRefIt (Lu et al., 2023) and Where2Place (Yuan et al., 2024b) benchmarks. We compared mainstream closed-source models and MLLM for enhancing spatial abilities (SpatialBot (Cai et al., 2024), SpaceLLaVA (Chen et al., 2024), and RoboBrain (Ji et al., 2025)). We used the proportion of predicted points within specified regions as the accuracy metric. For models without point output support, we asked models to output bounding boxes of target regions, then sampled evenly within these bounding boxes. *Spatial Visual Aids Capabilities:* We utilized our VABench to evaluate the capabilities. We found few models with this capability for Visual Trace prediction, so we also trained an end-to-end prediction baseline

**Table 2:** Performance comparison on object/free space reference benchmarks. The best results are highlighted.

| Benchmark | GPT-4o | SpaceLLaVA | LLaVA-NeXT-34B | SpatialBot-3B | ASMv2-13B | RoboBrain-7B | RoboPoint-13B | FSD-13B |
|---|---|---|---|---|---|---|---|---|
| RoboRefIt | 15.3 | 21.3 | 19.9 | 23.6 | 48.4 | 10.1 | 49.8 | **56.7** |
| Where2Place | 29.1 | 11.8 | 15.0 | 15.0 | 22.0 | 16.6 | **46.0** | 45.8 |

**Table 3: Performance comparison on VABench.** The best results are highlighted in bold.

**(a)** VABench-Point

| Model | Accuracy ↑ |
|---|---|
| GPT4o | 9.30 |
| ASMv2 | 10.07 |
| RoboPoint | 19.09 |
| RoboBrain | 7.00 |
| FSD | **61.82** |
|    w/o SrCoT | 26.21 |
|    w/o Alignment | 55.92 |

**(b)** VABench-VisualTrace

| Model | RMSE↓ | MAE↓ | LLM Score↑ |
|---|---|---|---|
| GPT4o | 136.13 | 113.53 | 4.37 |
| DINOv2 Predictor | 128.32 | 117.49 | 4.01 |
| RoboBrain | 121.6 | 103.8 | 4.5 |
| FSD | **78.26** | **63.44** | **6.21** |
|    w/o SrCoT | 99.53 | 80.06 | 5.07 |
|    w/o Alignment | 80.48 | 66.80 | 5.92 |

model using a pre-trained DINOv2 (Oquab et al., 2023) encoder coupled with transformer (Vaswani et al., 2017) architecture to predict visual trajectories, trained on the same visual trajectory data, named *DINOv2 Predictor*. We conducted this comparison to demonstrate the advantages of our reasoning-based FSD approach. More experimental details are provided in App. E.

**FSD exhibits superior general spatial reasoning capabilities.** As shown in Tab. 1, FSD achieves a leading average rank of 1.3 across 15 tasks from spatial benchmarks, significantly outperforming other 13B open-source models and rivaling the closed-source GPT-4o. The model particularly excels in 3D depth perception (88.0%), distance estimation (86.7%), and spatial relationship understanding (78.3%). These results validate our data-centric approach for building robust spatial reasoning, a critical foundation for advanced embodied intelligence.

**FSD excels in object reference and free space localization.** The results in Tab. 2 demonstrate FSD's ability to accurately identify objects and free spaces from language instructions. For object reference (RoboRefIt), FSD achieves 56.7% accuracy, surpassing both GPT-4o (15.3%) and specialist models like RoboPoint (49.8%) by significant margins. On the more challenging free space reference task (Where2Place), FSD performs competitively with RoboPoint while substantially outperforming other models. This improvement stems from enhanced spatial understanding through our SrCoT mechanism. See App. F for more visualizations.

**FSD demonstrates breakthrough performance in visual aid generation.** As shown in Tab. 3, FSD significantly outperforms all baselines in generating precise spatial affordances and visual traces. Specifically, FSD achieves 61.82% accuracy on VABench-P, over 3x higher than RoboPoint (19.09%) and attains significantly lower error rates with a better LLM Score on VABench-V. Ablation studies validate the critical contributions of both SrCoT and self-consistency alignment, confirming that our reasoning-based approach enables more accurate predictions than purely data-driven methods. A detailed comparison with RoboBrain (Ji et al., 2025) is available in App. I.

**Visualization of visual aids generated by FSD**. We visualize the visual aids generated by FSD in Fig. 5, showcasing its effective adaptation to diverse scenes, perspectives, and tasks. Fig. 6 illustrates how the CoT reasoning guides the generation process. Additional visualizations and complete output examples are available in App. F.

## 6.2 EVALUATION OF THE DECISION CAPABILITY

**SimplerEnv Evaluation.** To assess the zero-shot generalization capability of FSD, we deployed this approach on the WidowX robotic arm and conducted experiments using SimplerEnv (Li et al., 2024c). Since end-to-end VLA methods require fine-tuning, for fair comparison, we ensured that all end-to-end baseline methods were trained using the BridgeData Walke et al. (2023) dataset and evaluated under the same settings. We benchmarked mainstream VLAs, including end-to-end models (Octo, the $\pi_0$ series, OpenVLA), the modular-based method MOKA (Liu et al., 2024a), and the affordance-based RoboPoint (Yuan et al., 2024b). As shown in Tab. 4, although the latest

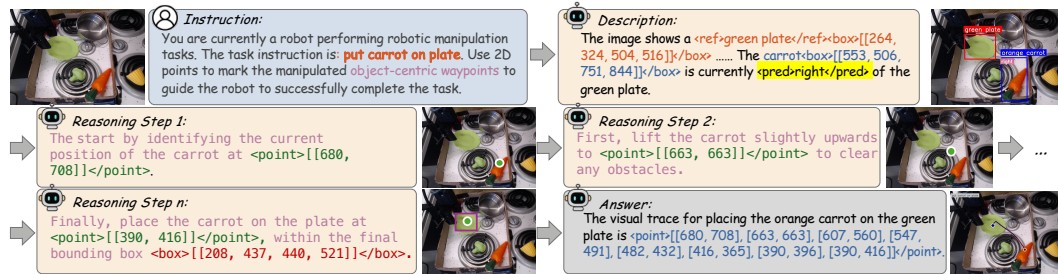

**Figure 6: Example of the FSD Reasoning Process** FSD performs point-by-point reasoning and localization, ultimately generating a feasible visual trace.

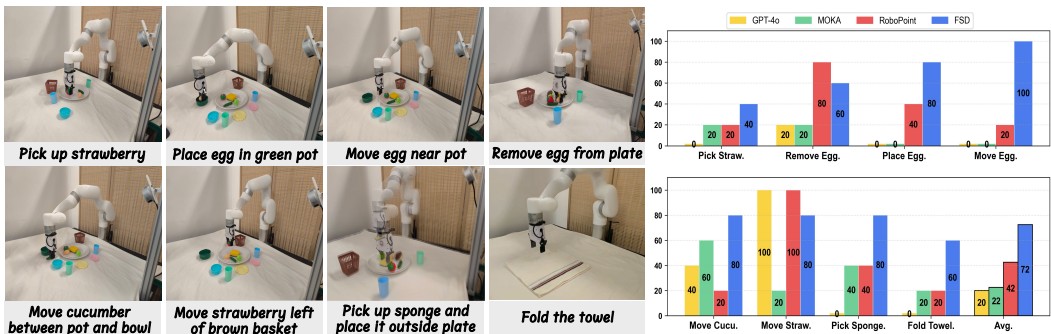

**Figure 7:** Real-world robotic manipulation tasks performed by an xArm 6 robot (left) and performance comparison of FSD against baseline models GPT-4o, MOKA, and RoboPoint (right).

**Table 4: SimplerEnv Evaluation on WidowX Robot.** Each task is tested 24 episodes. Most of the results for end-to-end VLAs are sourced from Chen et al. (2025), while the results for the remaining models are reproduced in accordance with the official code.

| Type | Model | Put Spoon on Towel | Put Carrot on Plate | Stack Green Block on Yellow Block | Put Eggplant in Yellow Basket | Avg |
|---|---|---|---|---|---|---|
| End-to-end VLA | Octo (Team et al., 2024) | 41.7 | 8.2 | 0.0 | 56.7 | 26.7 |
| | $\pi_0$ (Black et al., 2024) | 29.1 | 0.0 | 16.6 | 62.5 | 27.1 |
| | $\pi_0$-fast (Pertsch et al., 2025) | 29.1 | 21.9 | 10.8 | 66.6 | **48.3** |
| | OpenVLA (Kim et al., 2024) | 4.2 | 0.0 | 0.0 | 16.7 | 5.2 |
| | OpenVLA-OFT (Kim et al., 2025) | 34.2 | 30.0 | 30.0 | **72.5** | 41.8 |
| Modular VLA | MOKA (Liu et al., 2024a) | **45.8** | 41.6 | **33.3** | 12.5 | 33.3 |
| Affordance VLA | RoboPoint (Yuan et al., 2024b) | 16.7 | 20.8 | 8.3 | 25.0 | 17.7 |
| | FSD | 41.6 | **50.0** | 33.3 | 37.5 | 40.6 |

end-to-end VLAs (such as $\pi_0$-fast and OpenVLA-OFT) perform well in this limited generalization scenario, FSD still achieves an average success rate of 40.6%, significantly outperforming standard zero-shot baselines such as RoboPoint (17.7%). Without dedicated fine-tuning, end-to-end VLAs may suffer from severe performance breakdowns (with success rates approaching zero) when faced with substantial variations in backgrounds and instructions. In contrast, FSD demonstrates robust zero-shot generalization. Therefore, we suggest that an important direction for future research is to explore closed-loop policy VLAs explicitly guided by visual trajectories, in order to combine robust planning capabilities with precise execution abilities.

**Real World Robot Evaluation** As shown in Fig. 7, we conducted zero-shot tests with FSD on an xArm 6 robot across 8 tabletop manipulation tasks. The setup included an Intel RealSense L515 LiDAR camera. To test the capabilities of different visual aids, we used visual trace for the sponge and folding tasks, while affordance points were used for other tasks. We compared against GPT-4o, MOKA, and RoboPoint baselines. Robopoint often makes incorrect predictions in tasks involving spatial understanding. The primary cause of MOKA failures stems from the cascading errors of multiple submodules. Under zero-shot conditions, FSD achieved 72% success rate, outperforming the strongest baseline by more than 30%. Notably, FSD successfully completed complex tasks with

visual trace generation, e.g. towel folding, which was beyond baseline capabilities. Full results are presented in Fig. 7. We refer to App. G for detailed setup and visualization.

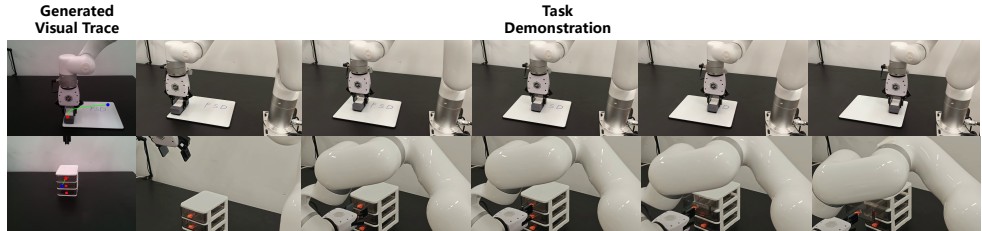

**Figure 8: Qualitative results on challenging manipulation scenarios**. We evaluate FSD on a contact-rich task (Top: Wiping the whiteboard) and an articulated object task (Bottom: Opening the drawer). The leftmost column visualizes the visual trace generated by FSD, followed by key frames of the real-world execution.

**Qualitative Experiments on Challenging Tasks** We conducted a preliminary exploration and attempted to assess FSD's capability in handling scenarios that require continuous contact and kinematic understanding. We introduced two new challenging tasks: "Wiping the whiteboard" and "opening the drawer." As shown in Fig. 8, FSD was able to infer the visual traces required for these tasks. For the contact-rich whiteboard task, the model generated a smooth wiping trajectory across the surface. For the articulated drawer task, the model successfully predicted the arc trajectory needed to pull the handle. Although the downstream action controller requires further adjustment and optimization for these more challenging tasks, these qualitative results verify the potential of FSD to bridge perception and complex control.

**Real World Execution Latency** We evaluated FSD's execution latency against the end-to-end OpenVLA and modular MOKA pipelines, with results reported in Tab. 5. FSD achieves the lowest latency due to its single-pass, single-model architecture that generates the entire visual trace in a zero-shot manner. This approach avoids both the costly

**Table 5:** Real-world Execution Time (s)

| Model | Sponge | Cucumber |
|---|---|---|
| OpenVLA (FT) | 23s | 12s |
| MOKA | 28s | 18s |
| **FSD** | **14s** | **10s** |

fine-tuning and step-by-step inference required by OpenVLA and the system overhead of a multi-component pipeline like MOKA. FSD thus effectively balances powerful zero-shot manipulation capabilities with practical, low real-world latency. We primarily use open-loop control, but FSD can also be integrated into a hierarchical learning-based pipeline, using its visual trace to guide and improve closed-loop control. See App. K for more details.

## 7 CONCLUSION

In this paper, we introduced FSD (*From Seeing to Doing*), bridging visual reasoning and robotic manipulation through intermediate spatial representations. Our approach overcomes the critical challenges of scarcity and heterogeneity through three key innovations: a Spatial Relationship-Focused Visual Chain-of-Thought for multi-step reasoning, a hierarchical weak-to-strong data pipeline, and a self-consistency mechanism to align spatial coordinates with visual signals. Experiments demonstrate FSD's superior performance across multiple spatial reasoning and visual aid benchmarks. In zero-shot robotic deployment, FSD achieved an impressive 72% success rate across diverse tasks, outperforming baselines by 30%. We acknowledge limitations, such as the reliance on 2D trajectory generation and constraints from training data quality. More limitations and future works are in App.J.

## ACKNOWLEDGEMENT

This work is supported by the National Natural Science Foundation of China (Grant Nos. 62422605, 62533021) and the National Natural Science Foundation of China Youth Student Basic Research Project (Grant No. 625B2128). Yifu Yuan is also supported by the Young Science and Technology Scientists Sponsorship Program by CAST - Doctoral Student Special Plan. This work is also supported by the National Natural Science Foundation of China (Grant No. 92370132) and the National Key Research and Development Program of China (Grant No. 2024YFE0210900). Pengyi Li is supported by the Basic Research Project (Grant No. 624B2101). We would like to thank

Zhongwen Xu, Liang Wang, Shuyang Gu, and Han Hu for their participation in the discussions of this paper and for providing valuable insights. In addition, we would especially like to thank Yiyang Huang for the constructive suggestions on improving the figures in the manuscript.

## ETHICS STATEMENT

This paper is dedicated to advancing the field of robotic manipulation towards the creation of more effective and versatile robotic assistants. Our research strictly adheres to responsible practices and aligns with the ICLR Code of Ethics. All training data was sourced from large-scale, open-access robotics datasets, with all assets utilized in full compliance with their original licensing and terms of service. We recognize that while the intended applications of this research are positive, the long-term societal impacts of increasing robotic autonomy warrant careful consideration.

## REPRODUCIBILITY STATEMENT

To promote transparency and reproducibility within the scientific community, we provide detailed training parameters and resources in the appendix. The complete code for both training and inference has been uploaded to an gitHub repository: https://github.com/pickxiguapi/Embodied-FSD. We have made datasets and model checkpoints publicly available.

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

## A WEAK-TO-STRONG DATASET CONSTRUCTION

**Region Grounding Dataset Generation** Regional grounding enables robots to focus on key objects within a scene. In embodied AI, robots need to concentrate on task-relevant local scenes in the image according to task instructions and be able to accurately locate objects. In Level 1 of our dataset, we integrate grounding with captioning, aiming for the agent to provide positional information when describing images. Unlike traditional image captioning, objects in embodied scenes are often cluttered, and we only seek to extract captions for task-relevant regions, avoiding redundant information. We prompt GPT-4o to exclude task-irrelevant regions based on task instructions and generate both image captions and the corresponding object names. Subsequently, we use Grounding DINO (Liu et al., 2023b) to capture the locations of various objects in the image and embed them into the caption sentences, forming captions with object location information. Below is an example from the dataset:

---

**Level 1: Region Grounding**

**Human:** Give a short and clear explanation of the subsequent image. **GPT:** A <ref>robotic arm</ref><box>[[338, 126, 994, 861]]</box> is positioned above a <ref>blue bottle</ref><box>[[257, 201, 381, 413]]</box>, with a <ref>grey toy</ref><box>[[391, 413, 538, 518]]</box> to the left and a <ref>green oval object</ref><box>[[592, 481, 702, 601]]</box> to the right on a wooden table.

---

**Spatial Relationship Dataset Generation** To accurately infer object spatial relationships from RGB images, a multi-stage pipeline is employed: encompassing object detection, instance segmentation, 2D-to-3D mapping, and subsequent relationship calculation.

Initially, for pre-identified objects of interest within the scene, GroundedSAM (Ren et al., 2024) is utilized to perform instance segmentation, yielding precise object masks. Subsequently, the 2D RGB information is elevated to a 3D spatial representation. This transformation begins with leveraging PerspectiveFields (Jin et al., 2023) to estimate the Z-axis orientation, serving as a coarse approximation for camera extrinsics. Concurrently, the WildCamera (Zhu et al., 2024) model is employed to estimate intrinsic camera parameters, including focal length and resolution. Metric3Dv2 (Hu et al., 2024) is then used for robust depth estimation. By integrating the RGB image, estimated depth image, and computed camera intrinsics and extrinsics, the 2D RGB image is transformed into a 3D point cloud. Given the prior object segmentation, the specific 3D spatial position and size of each individual object can be precisely derived from the generated point cloud. These comprehensive 3D data then enable the calculation of relative spatial relationships between objects, which are subsequently exported as a spatial relationship graph. We present the 2D RGB-D images before transformation and the visualized point clouds after transformation in Fig. 9.

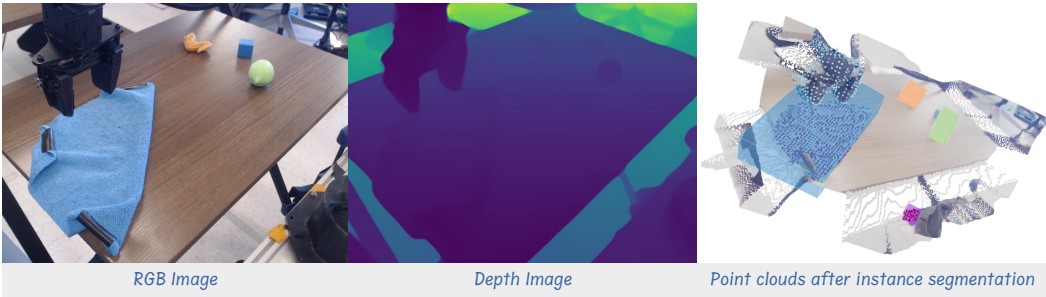

|RGB Image | Depth Image | Point clouds after instance segmentation|

**Figure 9:** RGB-D images before transformation and the visualized point clouds after transformation.

It is important to note that for inferring spatial relationships, only relative depth sorting data is generated, which means stringent accuracy requirements for absolute depth are relaxed. To further enhance data quality, particularly for robust relationship inference, objects exhibiting a relative depth gap of at least 20% are preferentially selected for subsequent generation. An example from the dataset:

---

**Level 2: Spatial Relationship**

```
Human:  Tell me about the items in your view and their
relationships to each other.  Answer the question with scene
graph.
GPT: <ref>front left burner</ref><box>[[597, 422,
817, 596]]</box> is positioned <pred>to the right
of</pred><box>[[597, 422, 817, 596]]</box><box>[[171, 275,
446, 409]]</box> <ref>metal pot</ref><box>[[171, 275, 446,
409]]</box>.  <ref>metal pot</ref><box>[[171, 275, 446,
409]]</box> appears <pred>to the left of</pred><box>[[171,
275, 446, 409]]</box><box>[[431, 474, 715, 608]]</box>
<ref>green spatula</ref><box>[[431, 474, 715, 608]]</box>.
<ref>metal pot</ref><box>[[171, 275, 446, 409]]</box> can
be seen <pred>to the left of</pred><box>[[171, 275, 446,
409]]</box><box>[[519, 408, 600, 490]]</box> <ref>orange
cheese slice</ref><box>[[519, 408, 600, 490]]</box>.
```

---

**Spatial Reasoning QA Generation** After generating the spatial relation diagrams, we can easily create various template questions for spatial reasoning (Cheng et al., 2024; Du et al., 2024), such as: *How are [A] and [B] positioned in relation to each other in the image? or From your perspective, which object in the image is at the shortest distance?* In addition to template-based QA, we also combine task instructions, existing spatial information, and images to query GPT-4o, thereby generating more diverse multi-turn dialogues to enhance the model's generalization ability in spatial reasoning.

**Spatial Affordance and Visual Trace Dataset Generation** Next, we provide a detailed description of how to extract the required controllable points/boxes and visual trajectories from the embodied dataset like BridgeData (Walke et al., 2023), RT-X (O'Neill et al., 2023) and Droid (Khazatsky et al., 2024). Our methodology for extracting visual aids, specifically Affordance Boxes, Affordance Points, and Visual Traces, involves a multi-stage process leveraging state-of-the-art vision models. We also incorporate a rigorous data validation procedure to ensure high-quality output.

First, we acquire the initial and final frames of the video sequence. To determine the Affordance Box, we utilize GroundingDINO (Liu et al., 2023b) and GroundedSam (Ren et al., 2024) to detect the mask of the `manipulated_object` in the final frame. The bounding box of this detected object in the final frame defines the Affordance Box, representing the ultimate spatial location of the manipulated object. Subsequently, we extract Affordance Points. This involves performing an erosion operation on the mask of the `manipulated_object` obtained from the final frame. Erosion reduces the mask's area, facilitating the sampling of points that are more central or internal to the object, thereby mitigating the risk of sampling edge points. From this eroded mask, we uniformly sample 8 points, which constitute the Affordance Points. For Visual Trace extraction, we begin by detecting the mask of the `manipulated_object` in the first frame. From this mask, we sample 3 points, which serve as the initial query points for the CoTracker (Karaev et al., 2024) model. Selecting multiple points enhances the robustness of trajectory tracking. The processed video sequence and these sampled points are then fed into the CoTracker model, which outputs the predicted trajectory for each query point across every frame of the video sequence. We calculate the total distance of each trajectory and select the longest trajectory as the representative trajectory for the `manipulated_object`. This chosen trajectory is then subjected to cubic spline interpolation for smoothing. Finally, 8 equidistant points are uniformly sampled from the smoothed trajectory, forming the Visual Trace.

A critical aspect of this process is addressing potential prediction errors from the pre-trained visual models, such as incorrect object identification or incomplete tracking of object motion. To mitigate these issues, we implement stringent rule-based filtering using hyperparameters like size thresholds and trajectory length thresholds. Before annotating each dataset, we iteratively adjust these rules and conduct manual inspections on 100 examples. Only when the accuracy of these filtered results surpasses 95% do we deem the filtering rules robust enough to proceed with the full data generation pipeline. This meticulous validation process ensures the high quality of the generated data.

Following the generation of these visual aids, we also pre-generate the thinking processes for CoT reasoning. We input templates, questions, and answers into GPT-4o, querying it to complete the thinking process. The complete query prompt is constructed accordingly:

---

**Prompt and In-context Example for Completing Thinking Process**

```
You are an AI visual assistant that can analyze a single
image.  You receive one image and corresponding caption,
task instruction, manipulated object in the task and target
place to finish the task, and bounding box position and
relative position of these objects.  Also, you receive the
answer of points and bbox.  Now, you need to generate a
<Description>...</Description>, <Reasoning>...</Reasoning>,
<Answer>...</Answer> format.
<Description> First, using the provided caption, task
instruction, describe the scene.  If there are errors in
the caption, please ignore them and do not point them out
in your description.  Instead of directly mentioning the
bounding box coordinates, utilize this data to explain
the scene using natural language with its bounding
box in the format like "<ref>object</ref><box>[[x1,
y1, x2, y2]]</box>".  When mentioning the predicate
between two objects, you should mention it in the
format like "<pred>predicate</pred><box>[[x1, y1, x2,
y2]]</box><box>[[x3, y3, x4, y4]]</box>", where "<box>[[x1,
y1, x2, y2]]</box>" denotes the bounding box coordinates of
the subject and "<box>[[x3, y3, x4, y4]]</box>" denotes the
bounding box coordinates of the object.  </Description>
<Reasoning> According to the task instruction and the answer
of points and bbox, provide a chain-of-thought, logical
explanation of the problem.  </Reasoning>
<Answer> State the final bounding box answer and point answer
in a clear and direct format here.  Bounding box answer is
in the format like "<box>[[x1, y1, x2, y2]]</box>".  Point
answer is in the format like "<point>[[x1, y1], [x2, y2],
[x3, y3], ...]</point>".  </Answer>
Here is an example:
Caption:  7up can is on the table beside the bowl, drawer is
open.
Task instruction:  place 7up can into top drawer
Manipulated object:  7up can<box>[[686, 174, 788, 357]]</box>
Target place:  top drawer <box>[[135, 411, 852, 920]]</box>
Current Relation:  (7up can, [686, 174, 788, 357], top
drawer, [135, 411, 852, 920], above)
Bounding Box Answer:  [569, 553, 651, 679]
Point Answer:  [[603, 609], [593, 637], [596, 643], [615,
596], [612, 593], [590, 637], [603, 606], [615, 631]]
<Description> The scene describes a <ref>7up
can</ref><box>[[686, 174, 788, 357]]</box> that is positioned
on a table.  It is situated <pred>above</pred><box>[[686,
174, 788, 357]]</box><box>[[135, 411, 852, 920]]</box> the
<ref>top drawer</ref><box>[[135, 411, 852, 920]]</box> which
is open.  </Description> <Reasoning> The 7up can is above the
top drawer now and should be placed into the top drawer.  So
the target position of the 7up can might be similar to the
position of the drawer's interior.  </Reasoning> <Answer>
To complete the task, the target position is <box>[[569,
553, 651, 679]]</box>.  The new position can also be roughly
defined by the following several free points <point>[[603,
609], [593, 637], [596, 643], [615, 596], [612, 593], [590,
637], [603, 606], [615, 631]]</point>, which can outline the
position of the 7up can after it is placed in the top drawer.
</Answer>
```

Next, we present two examples from the dataset: one involves generating Affordance Boxes/Points (Level 4), and the other involves generating visual trace (Level 5). We consider Affordance Points as samples within the box; thus, generating the box first and then the points is a refinement process from coarse to fine granularity. Therefore, both types are generated for each data, and one can choose which to use as needed. In the task of generating visual trace, since we performed equidistant interpolation in advance, the visual trace in the dataset is fixed at 8 points. Consequently, models trained on such a dataset will also generate 8 points as the visual trace.

---

**Level 4: Spatial Affordance Box/Points Generation**

```
Human:  "You are currently a robot performing robotic
manipulation tasks.  Your task instruction:  Moved the can
to the left side of the green cloth.  Observe the image, use
2D points and bounding box to mark the target location where
the manipulated object will be moved.  In your answer, use
<box>[[x1, y1, x2, y2]]</box> to present the bounding box
of the target region, and use <point>[[x1, y1], [x2, y2],
...]</point> to mark the points of the free space."

GPT: "<Description>
The image shows a <ref>white soup can</ref><box>[[544, 177,
700, 368]]</box> positioned to the right of a <ref>green
towel</ref><box>[[244, 389, 664, 676]]</box> on a table.  A
small bowl containing a yellow object and a brush is next to
the towel.
</Description>
<Reasoning>
The task is to move the white soup can to the left side of
the green towel.  Currently, the can is to the right of the
towel, so it needs to be repositioned to the left of the
towel's current location.
</Reasoning>
<Answer>
The target position for the white soup can is <box>[[250,
181, 400, 392]]</box>.  The new position can also be roughly
defined by the following points:  <point>[[346, 248], [302,
365], [377, 251], [330, 295], [357, 291], [354, 362], [329,
355], [312, 352]]</point>.
</Answer>"
```

---

---

**Level 5: Visual Trace Generation**

```
Human:  "You are currently a robot performing robotic
manipulation tasks.  Your task instruction:  put the green
spatula in the silver pot.  Observe the image, use 2D points
to mark the manipulated object-centric waypoints to guide the
robot to manipulate the object.  Typically, the waypoints
consist of an ordered sequence of eight 2D points.  The
format is <point>[[x1, y1], [x2, y2], ...]</point>."

GPT: "<Description>
The image shows a <ref>green spatula</ref><box>[[762, 536,
856, 711]]</box> positioned to the right of an orange towel
on a table.  A <ref>silver pot</ref><box>[[0, 723, 242,
872]]</box> is located above a robotic arm.  Other objects,
such as a gray can and a mushroom, are also present in the
scene.
</Description>
<Reasoning>
To move the green spatula into the silver pot, start
by identifying the initial position of the spatula at
<point>[[802, 613]]</point>.  First, lift the spatula upwards
to clear the table, moving to <point>[[774, 501]]</point>.
Then, move it horizontally towards the left, passing through
<point>[[685, 394]]</point>.  Finally, lower the spatula into
the silver pot, ending at <point>[[657, 401]]</point> within
the target bounding box <box>[[225, 529, 407, 640]]</box>.
</Reasoning>
<Answer>
The visual trace for placing the green spatula into the
silver pot is <point>[[802, 613], [780, 582], [774, 501],
[744, 465], [685, 394], [657, 349], [668, 354], [657,
401]]</point>.
</Answer>"
```

## B  TRAINING DETAILS AND DATASETS

The training of FSD is a two-stage process, building upon the LLaVA (Liu et al., 2024b) architecture through continued fine-tuning of ASMv2 (Wang et al., 2025). In order to train a VLM with powerful spatiotemporal reasoning abilities that can also generate visual aids, we leveraged a comprehensive dataset of approximately 1.4M samples from various sources. *Note:* The position coordinate format used by FSD normalizes the image coordinates to a range of 0-999 after padding the image to a square shape. For all datasets mentioned below, the coordinates have been pre-processed in this manner.

**Stage 1: Spatial Reasoning Enhancement** The primary goal of the first stage is to enhance the model's spatial reasoning capabilities. This dataset is primarily composed of two parts: common-sense image QA and conversations data and spatial reasoning data. The inclusion of GeneralQA is crucial for FSD to maintain broad instruction-following abilities after fine-tuning. First, we selected some datasets and randomly sampled around 838k data from LLaVA-665k and ASMv2-4M instruction-following data. Next, we incorporated approximately 295k samples of spatial reasoning data from the LLaVA-OneVision (Li et al., 2024a), RoboPoint (Yuan et al., 2024b), SpatialBot (Cai et al., 2024) and SAT (Ray et al., 2024) training datasets. Finally, the first three levels of FSD's collected embodied spatial reasoning datasets (250k samples) were also integrated into this training phase. A summary of the datasets used can be found in Tab. 6.

**Stage 2: Visual Aids Generation and Understanding** We specifically focused on training for visual aids generation. Here, "generation" refers to generating visual aids based on a given question, while

**Table 6:** Details of the training data for FSD

| Stage | Task | Datasets | Samples | Data Sources |
|---|---|---|---|---|
| Stage 1 | GeneralQA (Caption, VQA, OCR, RegionVQA, Conversation, Grounding, Text) | ShareGPT4V, VQAv2, OCR-VQA, Visual7W, ST-VQA, RefCOCO/+/g, VG, AS-Core, AS-V2, TextVQA, Visual7W | 838k | LLaVA (Liu et al., 2024b), ASMv2 (Wang et al., 2025) |
| | General Spatial Reasoning | KITTI, 2D3DS, ObjectRef, RegionRef, VSR, CLEVR, CLEVR-Math, SUPER-CLEVR, RAVEN | 295k | SpatialBot (Cai et al., 2024), RoboPoint (Yuan et al., 2024b), SAT (Ray et al., 2024), LLaVA-OneVision (Li et al., 2024a) |
| | Embodied Spatial Reasoning | FSD Level 1 (145k), Level 2 (86k) and Level 3 (19k) | 250k | FSD |
| Stage 2 | Spatial Affordance Generation & Understanding | FSD Level 4 (24k) | 24k | FSD |
| | Visual Trace Generation & Understanding | FSD Level 5 (26k) | 26k | FSD |

"understanding" represents its inverse problem. We excluded some inverse problems with ambiguous semantics. The final total amount of data used is shown in Tab. 6.

**Training Configuration** We used exactly the same hyperparameters in both stages of training. We utilized a global batch size of 128 and the AdamW optimizer, configured with $\beta_1 = 0.9$, $\beta_2 = 0.999$, and a weight decay coefficient of 0. The learning rate was set to $2 \times 10^{-5}$. Both training stages employ a linear warmup for the first 3% of training steps, followed by a cosine decay strategy to a minimum learning rate of 0. We simultaneously train both the vision-language connector and the language model. The image resolution was set to $336 \times 336$, and the visual encoder remained frozen throughout the entire training process. In the first phase, we train for 1 epoch on the complete dataset, while in the second phase, we train for 3 epochs. The FSD model has approximately 13B trainable parameters and we conducted training using 8 A100 40G GPUs, with Stage 1 requiring approximately 72 hours and Stage 2 requiring 8 hours.

## C  DETAILS OF ACTION EXECUTION

When utilizing FSD for robotic manipulation tasks, we can select from various visual aids. As described in Fig. 2, these include **spatial affordance boxes** ($\mathcal{B}$), **spatial affordance points** ($\mathcal{P}$), and **object-centric visual traces** ($\tau$). The choice of visual aids dictates the subsequent motion planning strategy.

**Motion Planning with Spatial Affordances** For spatial affordance boxes ($\mathcal{B}$), the target point for manipulation is derived by sampling the center of the box. In the case of spatial affordance points ($\mathcal{P}$), a point is directly sampled. When relying on spatial affordance information, whether from boxes or points, we employ CuRobo (Sundaralingam et al., 2023) as our motion planner. CuRobo is responsible for generating collision-free paths that guide the robot's end-effector to the inferred target affordance point.

**Motion Planning with Object-Centric Visual Traces** When leveraging object-centric visual traces ($\tau$), the process involves mapping 2D visual traces into 3D space and then interpolating these discrete points to form a complete motion trajectory in SE(3) space. The detailed procedure is as follows:

We directly acquire 2D keypoint information, denoted as $k_i = (u_i, v_i) \in \mathbb{R}^2$, where $u_i$ and $v_i$ represent the x and y coordinates of the $i$-th point in the image, for $i \in [1, T]$. Initial depth information, $d_i \in \mathbb{R}$, is obtained from a depth camera. Using the Pinhole camera model (Hartley & Zisserman, 2004), we can transform these 2D keypoints into 3D Cartesian coordinates $P_i = (x_i, y_i, z_i) \in \mathbb{R}^3$ via:

$$s_i \begin{bmatrix} u_i \\ v_i \\ 1 \end{bmatrix} = \begin{bmatrix} f_x & 0 & c_x \\ 0 & f_y & c_y \\ 0 & 0 & 1 \end{bmatrix} \begin{bmatrix} x_i \\ y_i \\ z_i \end{bmatrix}$$

Here, $s_i$ is the normalized depth, calculated as $s_i = d_i/\texttt{depth\_scale}$. The intrinsic camera parameters, $f_x, f_y, c_x, c_y$, and the depth scaling factor, $\texttt{depth\_scale}$, are all camera-specific.

However, a naive use of the raw depth values $d_i$ obtained from the depth camera often results in trajectories that closely hug the object's surface, which is undesirable for robust robot manipulation. To address this, we formulate an optimization problem to address it. We fix the depth values for the start and end points of the path ($d_1$ and $d_T$) and optimize the intermediate depth values $d_{2:T-1}$ to minimize the total Euclidean distance between consecutive points in Cartesian space.

The objective function for this optimization is:

$$\hat{d}_i = \arg \min_{d_{2:T-1}} \sum_i d(P_i, P_{i+1})$$

where $d(P_i, P_{i+1})$ represents the Euclidean distance between points $P_i$ and $P_{i+1}$. We employ a gradient descent method from $\texttt{scipy}$ library to optimize this objective function. This refined approach allows for more robust and practical robot motion planning by addressing the limitations of raw depth data and providing a structured framework for integrating various visual aids.

## D  DETAILS OF VISUAL AIDS GENERATION BENCHMARK

To address the gap in evaluating visual assistance signal generation, we established **VABench**, a comprehensive benchmark. VABench comprises two distinct tasks: **VABench-Point** and **VABench-VisualTrace**, each featuring 300 meticulously hand-annotated questions. These tasks require models to infer visual auxiliary information solely from natural language instructions, mimicking everyday human commands.

For **VABench-Point**, we provide ground truth bounding boxes for each question. The model's performance is then assessed by calculating the proportion of predicted points that fall within the target region. For models that only output bounding boxes, we explored two scoring methods: Intersection Over Union (IOU) and uniformly sampling points within the predicted box. We ultimately opted for the latter approach, as the physical interaction between a robotic arm and an object in real-world tasks is fundamentally determined by specific point coordinates. For **VABench-VisualTrace**, we provide ground truth trajectories consisting of eight points. When the predicted trajectory length deviates from the ground truth, we employ interpolation to align their lengths. To ensure consistent evaluation across varying image resolutions, all coordinates are uniformly normalized to a range of 0 to 1000. Subsequently, we employ a combination of Mean Absolute Error (MAE), Root Mean Squared Error (RMSE), and **GPT Score** to comprehensively evaluate performance.

Here, we detail the design philosophy and evaluation procedure of the GPT Score. Given that multiple valid solutions can exist for each task instruction, relying solely on trajectory similarity to the ground truth as a scoring criterion can be one-sided. Therefore, we established detailed evaluation criteria to simulate realistic human assessment. Based on these rules, we introduced a visualized trajectory scoring method leveraging Multimodal Large Language Models (MLLMs), termed **GPT-score**.

Specifically, we designed an evaluation prompt to guide GPT-4.1 in assessing predicted object manipulation trajectories based on both task instructions and visual inputs. The prompt provides clear instructions and criteria, positioning the model as an expert evaluator in robotic manipulation and visual reasoning. Each evaluation instance consists of a task instruction and an accompanying image that visualizes the predicted trajectory, where a red circle indicates the start point and a blue diamond marks the end point. The model is instructed to assess the trajectory according to three key criteria: (1) task alignment and success, determining whether the predicted path correctly fulfills the instruction by starting and ending in appropriate locations; (2) feasibility, evaluating the physical plausibility and smoothness of the motion; and (3) obstacle avoidance, considering whether the trajectory avoids potential collisions. The prompt emphasizes that completing the task correctly is the most important

factor; any major deviation in goal achievement results in a low score, even if the trajectory appears smooth or feasible. Then, the model returns a structured response consisting of a numerical score from 1 to 10, along with a concise explanation. Scores are interpreted based on task success and quality: low scores (1–4) indicate failure to accomplish the task, mid-range scores (6–8) reflect successful but imperfect trajectories, and high scores (9–10) are reserved for trajectories that are both accurate and high-quality. This scoring scheme allows for nuanced, human-like evaluation that integrates both semantic understanding and visual reasoning. By leveraging this multimodal prompt framework, GPT-score enables a robust and interpretable evaluation process that aligns closely with human judgment, overcoming the limitations of purely geometric or distance-based metrics. The complete prompt is presented as follows:

---

**Complete Prompt for GPT Score Metrics**

You are an expert evaluator in robotic manipulation and visual reasoning. Your job is to assess the quality of predicted trajectories based on task instructions and visual inputs.

You are given:
- A task instruction describing an object manipulation task.
- An image showing a predicted trajectory.

**Note:**
- In the image, the red circle indicates the start point, and the blue diamond indicates the end point.
- The trajectory represents the predicted movement path of the manipulated object, not the robot or end-effector.
- You should **evaluate the predicted trajectory as a proposed motion for the object that is supposed to be moved**, based on the task instruction — **not based on the static positions of objects in the image**. The objects have not actually moved.

**Evaluation Criteria (listed in order of importance):**
1. **Task Alignment and Success (most important)** - Does the trajectory clearly and accurately fulfill the task instruction? - **The trajectory must start at the correct location and end at a target position that aligns with the task goal.** - Large deviations in the starting or ending point (e.g., wrong object, wrong destination, or stopping short of the goal) should result in a low score, even if the rest of the trajectory is smooth. - If the task is not accomplished (due to incorrect goal interpretation or spatial execution), the score should be low regardless of other qualities.
2. **Feasibility** - Is the movement physically plausible, smooth, and continuous? - Are there any unrealistic discontinuities, sharp turns, or impossible transitions? - Even if the movement is feasible, it should not receive a high score if the task is not completed.
3. **Obstacle Avoidance / Safety** - Does the trajectory reasonably avoid collisions with surrounding objects? - Minor risks may be tolerated if the task is completed successfully, but major or clear collisions should reduce the score.

**Scoring Guideline:**
- If the task is **not accomplished**, or if the start or end point is significantly incorrect, the score should typically be **4 or below**.
- If the task is completed but the trajectory has issues (e.g., roughness, minor risk of collision), a score in the **6–8** range is appropriate.
- A **score of 9–10** should be given only when the trajectory clearly completes the task, with good start/end accuracy, smooth motion, and reasonable safety.

Based on these criteria, provide a single overall score from 1 (very poor) to 10 (excellent), reflecting how well the task is accomplished.

Respond strictly in the following format:
Score: <1-10>
Explanation: <brief justification>

The task instruction is: {task_instruction}

Please give your response.

---

# E    DETAILS OF BENCHMARKS AND BASELINES

For **general spatial reasoning tasks**, the answers are typically multiple-choice questions with clear options. However, some spatial reasoning models show reduced instruction-following ability after fine-tuning, preventing them from directly outputting the correct option. To address this, we use a

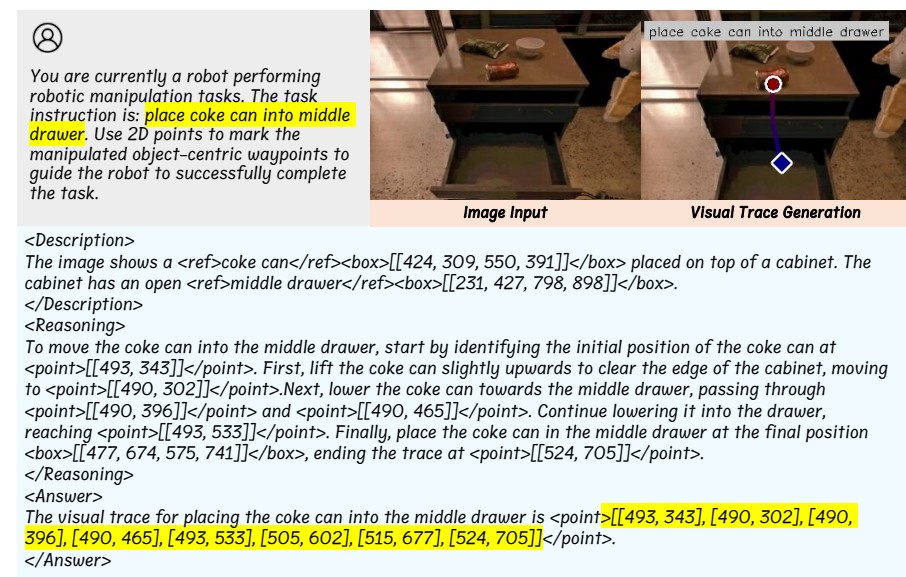

**Figure 10:** An example and visualization of FSD for generating visual trace.

**lenient matching rule**, considering an answer correct if it includes either the correct content or the corresponding option.

For **object/region reference tasks**, we carefully fine-tuned and used a tailored prompt for each model. Most models, such as GPT-4o and ASMV2 (Wang et al., 2025), cannot directly output specific points. Similar to the validation process for RoboPoint (Yuan et al., 2024b), we also found that using in-context learning to specify point output formats resulted in worse performance compared to directly outputting bounding boxes. Therefore, for these models, we instructed them to output bounding boxes directly. From these bounding boxes, we either uniformly sampled nine points or took the midpoint. Then, we calculated the proportion of points within the specified region to determine the final average accuracy.

For the **VABench-VisualTrace task**, due to a lack of strong baselines, we developed an additional Transformer-based prediction model, trained from scratch using the same data, which we named **DINOv2 Predictor**. In the DINOv2 Predictor, the visual encoder uses a pre-trained DINOv2 (Oquab et al., 2023), encoding images to output a $(196, 768)$ feature vector. The language encoder uses a pre-trained T5-Base (Raffel et al., 2020), outputting $(32, 768)$. These are concatenated with learnable embeddings $(8, 768)$ and passed through a Transformer encoder base. The output of the learnable embeddings is then read and passed through a linear layer to predict eight points. During training, we keep the language encoder fully frozen and train the visual encoder along with the other remaining parameters.

## F MORE VISUALIZATIONS AND EXAMPLES

The visualization of the output of FSD is shown in Fig. 10 and Fig. 11. Besides, we present the prediction results of FSD on Where2place (Yuan et al., 2024b), Roborefit (Lu et al., 2023), and VABench in Fig. 12, Fig. 13, and Fig. 14, respectively.

## G REAL WORLD EXPERIMENT RESULTS

In our real-world desktop manipulation tasks, we used an xArm 6 robotic arm for evaluation. This setup included an Intel RealSense L515 LiDAR camera and a force-torque sensor on the xArm to enable compliance control, which improved interaction with the environment. A computer running Ubuntu 24.04 and equipped with an NVIDIA GTX 1660 was directly connected to the arm and camera to execute low-level control policies. Notably, a single RealSense L515 depth camera was

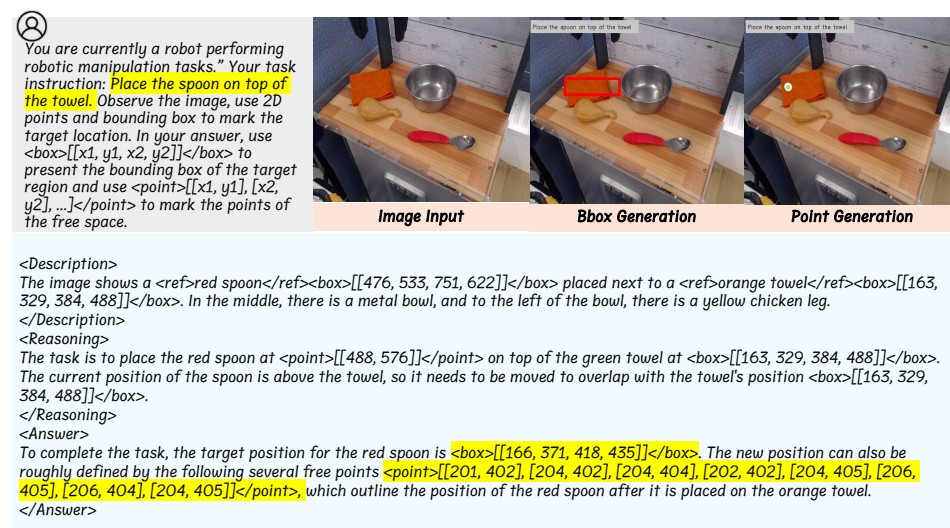

**Figure 11:** An example and visualization of FSD for generating affordance box and points.

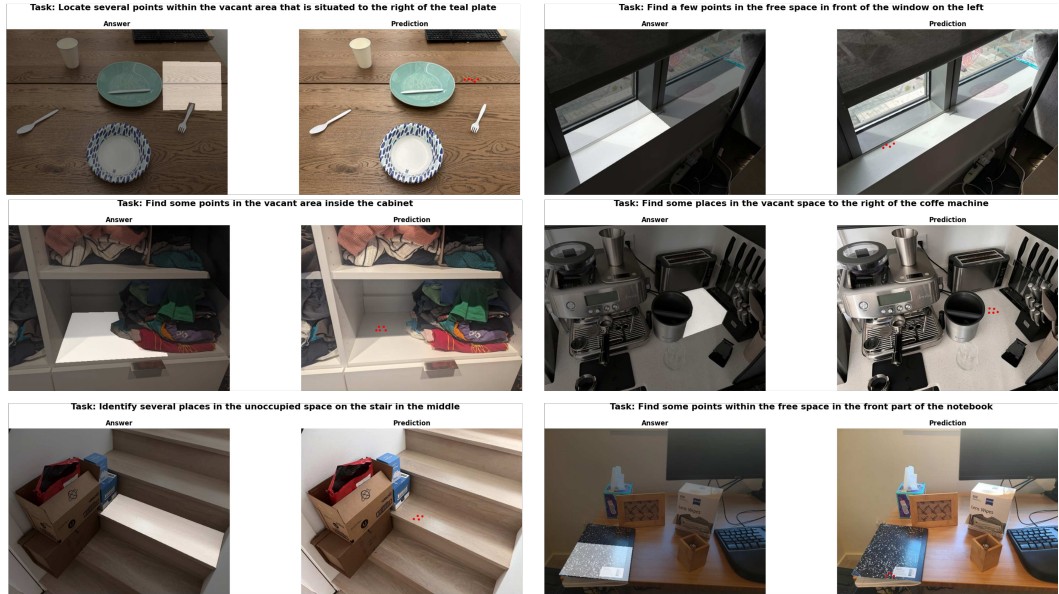

**Figure 12: Visualization of visual aids generated by FSD in the Where2Place benchmark.**

sufficient for task completion, especially when performing visual trace execution. This approach eliminated the need for object segmentation and 3D mapping; instead, we directly mapped 2D visual trajectories to 3D for execution, with no strict requirements on depth information accuracy. Demonstrations are available in Fig. 15 and on our website.

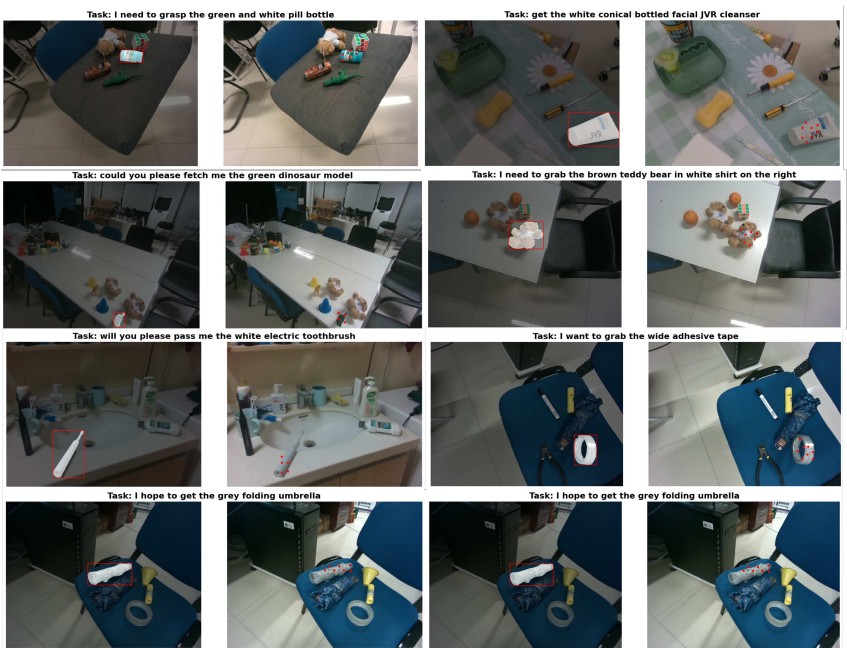

**Figure 13: Visualization of visual aids generated by FSD in the RoboRefit benchmark.**

## H   PROMPT FOR USING FSD MODEL

> **Generate Spatial Affordance Points & Bounding Box**
>
> You are currently a robot performing robotic manipulation tasks. Your task instruction: {Task Instruction}. Observe the image, use 2D points and bounding box to mark the target location where the manipulated object will be moved. In your answer, use <box>[[x1, y1, x2, y2]]</box> to present the bounding box of the target region, and use <point>[[x1, y1], [x2, y2], ...]</point> to mark the points of the free space.

> **Generate Visual Trace**
>
> You are currently a robot performing robotic manipulation tasks. Your task instruction: {Task Instruction}. Observe the image, use 2D points to mark the manipulated object-centric waypoints to guide the robot to manipulate the object.Typically, the waypoints consists of an ordered sequence of eight 2D points. The format is <point>[[x1, y1], [x2, y2], ...]</point>.

## I   COMPARISON OF FSD AND ROBOBRAIN

Both FSD and RoboBrain (Ji et al., 2025; Team et al., 2025) have the capability to generate visual trace. However, RoboBrain tends to produce agent-centric visual trace, whereas FSD generates object-centric visual trace. FSD adopts a task-centric design principle, allowing it to perform effectively even in more heterogeneous ontological scenarios, including those that completely lack robotic arms in the image, thus exhibiting stronger generalizability. Due to the differences in the methods of generating visual trace, we conducted several sets of visual trajectory visualizations for qualitative analysis, as shown in Fig. 16. Under the same zero-shot setting, the visual trace generated by FSD has higher accuracy compared to those generated by RoboBrain, confirming that FSD's reasoning-based pipeline possesses greater generalizability when facing unknown tasks.

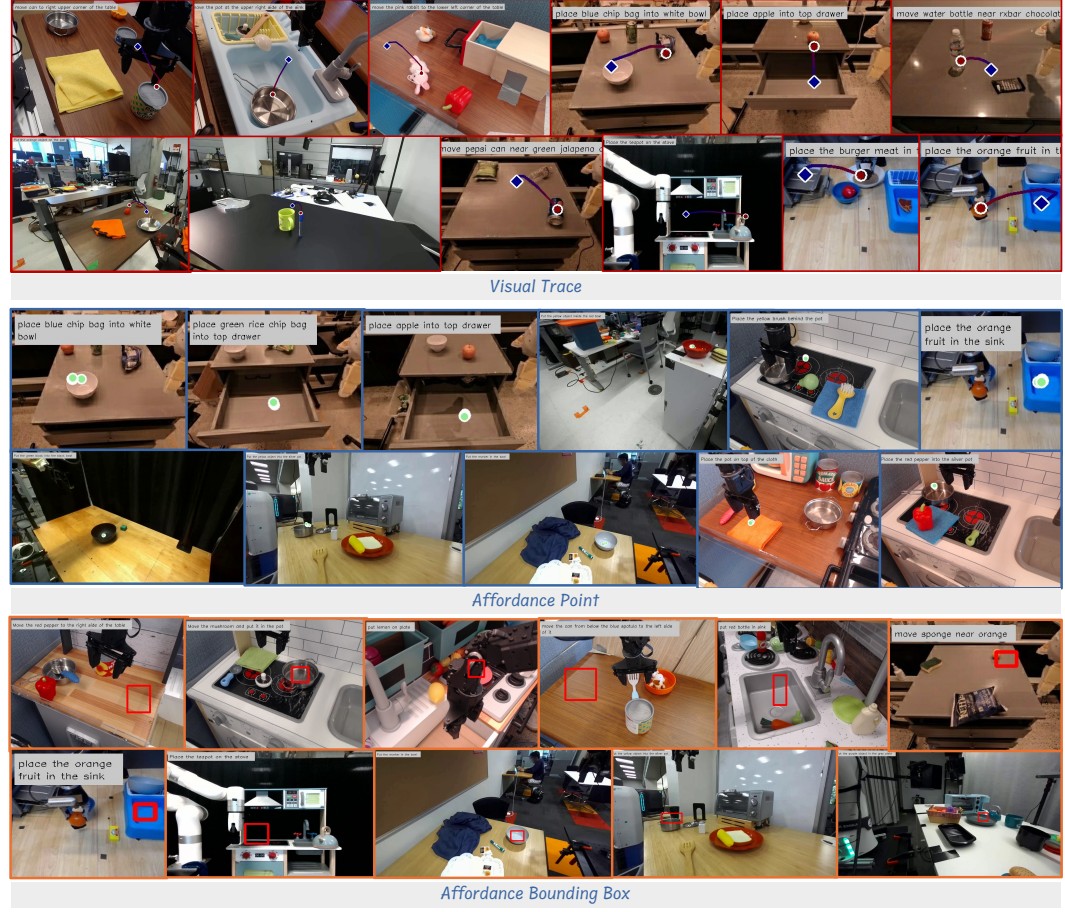

**Figure 14: Visualization of visual aids generated by FSD in the VABench benchmark.** FSD can generate three types of visual aids based on task instructions for novel tasks and scenarios. 1st-2nd row: visual trace; 3rd-4th row: affordance points; 5th-6th rows: affordance bounding box.

## J FUTURE WORKS

We have made preliminary attempts to use visual aids as intermediate states in FSD, achieving promising results in object/target/region reference and actual manipulation task execution. Future work can focus on the following aspects to further enhance the applicability of this paradigm: 1. **Task Decomposition for Complex and Long-Horizon Instructions:** The current version of FSD primarily targets clear and explicit language instructions. When dealing with long-horizon tasks or ambiguous/complex instructions, the model needs to decompose them into atomic, executable sub-tasks. We believe that decomposing instructions into a sequence of visual aids to guide each sub-task execution is a promising avenue. 2.**Downstream Execution and Visual-Aid-Guided Control:** Currently, FSD relies mainly on training-free motion planning methods for downstream execution. In extremely complex or dynamic scenarios, this may lead to a bottleneck in success rates. A potential improvement is to use the generated visual aids as explicit guidance for downstream VLA models, replacing language-conditioned training. Several preliminary studies (Zheng et al., 2024; Bharadhwaj et al., 2024; Li et al., 2025) have shown that, for robotic manipulation tasks, affordance and visual trajectories provide more effective guidance than language prompts. 3. **Extending from 2D to 3D Visual Aids:** Zhou et al. (2025b) At present, FSD focuses on predicting 2D visual aids, similar to the representation used in Referring Expression Comprehension (REC) and related tasks, which leverages the general visual understanding and reasoning capabilities of VLMs. However, as task and scene complexity increase, predicting 3D visual traces may prove to be a more effective solution, and we identify this as an important direction for future research. 4. **Inference Efficiency of FSD:** While FSD demonstrates acceptable efficiency for open-loop execution, the inference latency associated

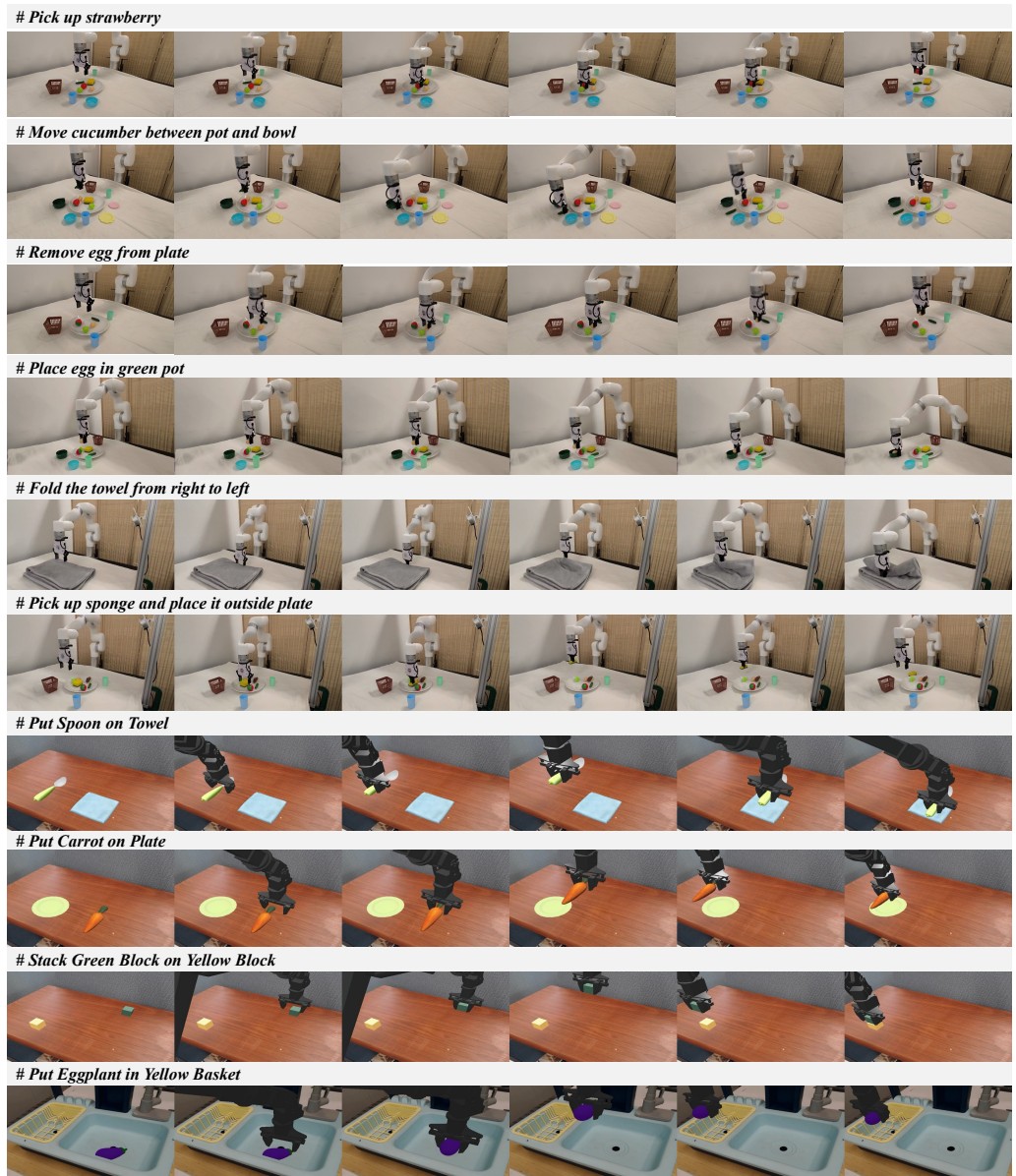

**Figure 15: Visualization of FSD executing tasks**. The first six rows are real-world experiments, and the last four rows are from SIMPLEREnv.

with its reasoning capabilities can become a bottleneck in dynamic environments, necessitating frequent re-planning. Consequently, optimizing inference speed to enable more responsive closed-loop adaptation remains a key priority for future work. 5. **Visual-Aid-Mediated Human Feedback and Alignment:** Traditional alignment techniques, such as Reinforcement Learning from Human Feedback (RLHF), often rely on language-based evaluations Yuan et al. (2024c); Liu et al. (2024d), which can be spatially imprecise and ambiguous for complex manipulation tasks. We envision using visual aids as a transparent and interactive interface for human intervention. This visual-aid-mediated feedback loop could significantly improve the sample efficiency of alignment and ensure that the agent's execution is more consistent with human intent and safety constraints.

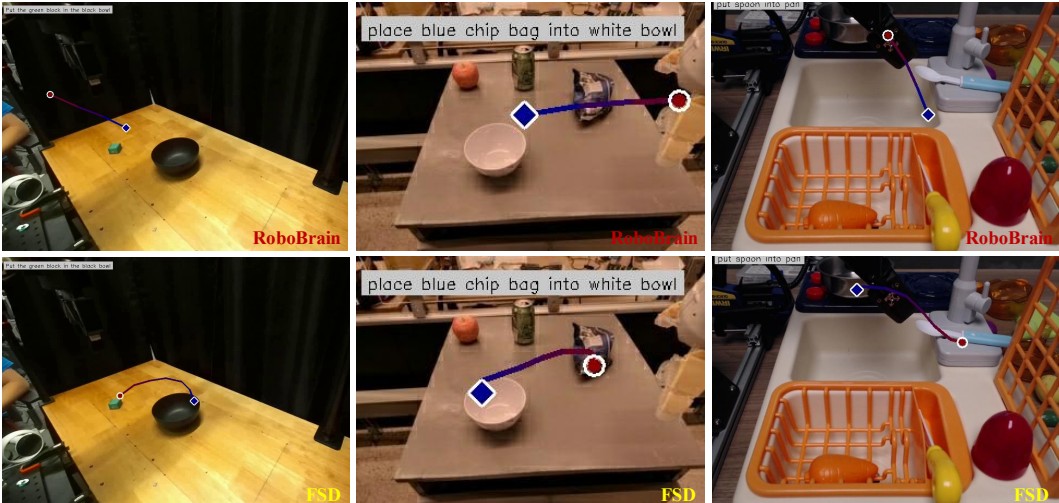

**Figure 16:** Comparison of generated visual traces between FSD and RoboBrain.

## K   INTEGRATION WITH LEARNING-BASED PIPELINE

**Experimental Setup.** To evaluate FSD as a complementary component for learning-based methods, we conducted experiments on 10 tasks from the LIBERO-spatial benchmark based on CleanDiffuser codebase Dong et al. (2024), each with 50 demonstrations. Our baseline is a Diffusion Policy (Chi et al., 2023) using a T5-base for text encoding. The policy is conditioned solely on the current RGB image to predict action chunks of length 16. For FSD integration, we adopt the ATM (Wen et al., 2023) approach: a visual trace is generated from the initial frame and task instruction, and we then train a trajectory-guided policy that leverages this visual guidance.

**Results and Analysis.** The results in Tab. 7 show that FSD improves the average success rate by nearly 10% over the baseline. We believe this is because the visual trace provides the policy with richer information by clearly indicating the object's movement path, thereby enhancing its ability to distinguish between different task instructions.

**Table 7:** Performance on the LIBERO-spatial benchmark. FSD provides a significant improvement when integrated with a Diffusion Policy (DP) baseline.

| Tasks | DP | DP + FSD |
|---|---|---|
| Libero Spatial (10 Tasks Avg.) | 76.1 ± 1.6 | 85.3 ± 1.2 |

## L   SUPPLEMENTARY EXPERIMENTS

### L.1   ABLATION ON HIERARCHY LEVELS

We designed the 5-level capability hierarchy as a logical weak-to-strong curriculum, where Levels 1-3 (Spatial Understanding) serve as prerequisites for Levels 4-5 (Visual Aid Generation). To validate the effectiveness of two-stage learning, we conducted a comparative experiment called "FSD w/o Stage 1". In this experiment, we skipped the foundational training (Levels 1-3) and instead trained the model directly on visual aid data (Levels 4-5). All other experimental settings remain unchanged compared to the FSD (Full).

The results are presented in Tab. 8. The results show that the FSD significantly outperforms the FSD w/o Stage 1 variant. The first stage provides essential spatial understanding and coordinate anchoring abilities. These are essential for the reasoning process and the generation of visual aids.

**Table 8: Ablation study on the impact of Stage 1 training.** We compare the full FSD model against a variant trained without the foundational spatial understanding stage (Levels 1-3).

| Model | Where2Place | VABench-Point |
|---|---|---|
| FSD (Full) | **45.8** | **61.8** |
| FSD w/o Stage 1 | 33.2 | 42.3 |

## L.2 ABLATION ON TRAJECTORY POINT DENSITY

In this section, we ablated the number of generated trajectory points. We fixed the model trained on Stage 1 data as the base model, and then adjusted the training data for Stage 2 using resampling to support the generation of $4, 8$, and $16$ points, respectively. We present the ablation results on VABench-V.

The comparative results are detailed in Tab. 9. The results indicate that using 4 points yields the lowest score due to the loss of prediction accuracy, while the prediction performance with 16 points slightly degrades and is inferior to that with 8 points. This may be attributed to the fact that a greater number of predicted points increases the difficulty of both instruction following and reasoning processes. We argue that more complex tasks require predicting additional points to characterize more intricate visual trajectories; however, for most tasks, 8 points can provide sufficient granularity to generate smooth trajectories, achieving a balance between performance and speed. Therefore, for the majority of manipulation tasks, we adopt $H = 8$ points as a design choice to balance granularity and token complexity.

**Table 9: Ablation results on VABench-V with different trajectory point densities.** We compare the performance metrics for trajectories generated with 4, 8, and 16 points.

| Method | RMSE ↓ | MAE ↓ | GPT Score ↑ |
|---|---|---|---|
| FSD-4p | 102.1 | 82.6 | 5.0 |
| FSD-8p (default) | **78.3** | **63.4** | **6.2** |
| FSD-16p | 85.2 | 67.1 | 5.7 |

## L.3 MORE COMPARISON ON REAL-WORLD TASKS

To further evaluate the robustness of our approach, we conducted additional comparative experiments with OpenVLA on two challenging real-world tasks using the xArm6 platform. Specifically, for OpenVLA, we report results after fine-tuning (FT) with 20 demonstrations per task, as its zero-shot success rate was 0%. In contrast, both MOKA and FSD were evaluated under zero-shot (ZS) conditions. The success rates for each method are summarized in the Tab. 10.

**Table 10: Comparison of success rates on real-world xArm6 tasks.** OpenVLA was fine-tuned (FT) with 20 demonstrations, while MOKA and FSD were evaluated in a zero-shot (ZS) setting.

| Task | OpenVLA (FT) | MOKA (ZS) | FSD (ZS) |
|---|---|---|---|
| Pick up sponge and place outside plate | **100%** | 40% | 80% |
| Move cucumber between pot and bowl | 40% | 60% | **80%** |

Notably, even after fine-tuning, OpenVLA (FT) performed poorly in the second task (40%), indicating inadequate generalization to positional variations within the test distribution. In contrast, FSD demonstrated consistent and robust performance (80%) across both tasks under zero-shot conditions. These results suggest that, compared to end-to-end methods, FSD's paradigm of reasoning-driven SrCoT, combined with intermediate visual aids, offers significant advantages in both generalization capability and execution accuracy.

### L.4 VISUALIZATION OF SELF-CONSISTENCY ALIGNMENT

To provide a more intuitive understanding of the impact of alignment, we present a qualitative case study in Fig. 17. As illustrated, the model without alignment (grey path) generates a correct textual reasoning plan but fails to map it to accurate coordinates, leading to "spatial hallucination." In contrast, the model with alignment (colored path) successfully anchors its reasoning to the correct pixel coordinates, ensuring precise task execution.

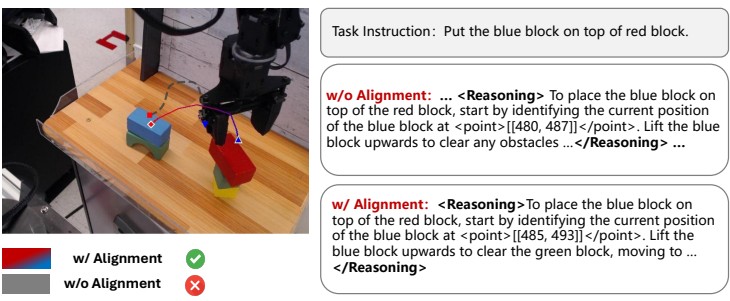

Figure 17: **Visual comparison demonstrating the effectiveness of Self-Consistency Alignment.**

It is worth noting that without self-consistent alignment, the model's textual reasoning process is logically correct (for example, it can accurately describe the target position relative to the red block). However, it fails to ground this reasoning spatially. The generated output coordinates deviate from the described location, resulting in a trajectory that does not align with physical reality. This indicates that without alignment, the model treats coordinates merely as abstract linguistic symbols, rather than mapping them to actual visual features, thereby leading to spatial hallucinations. In contrast, the proposed bidirectional training mechanism with self-consistency alignment enforces a strict correspondence between spatial coordinates and visual features. This compels the model to focus on specific pixels corresponding to the generated coordinates, ensuring that the generated points accurately land on the intended semantic targets and mitigating coordinate drift.

## M   USE OF LLMS

We utilized a Large Language Model (LLM), specifically, as a writing assistant during the preparation of this manuscript. The use of the LLM was strictly limited to language enhancement tasks, such as improving grammar, rephrasing sentences for better clarity, and ensuring stylistic consistency. The LLM was not used for generating scientific ideas, experimental results, or conclusions. All authors have carefully reviewed and revised the LLM's suggestions and take full responsibility for the scientific integrity and content of this work.

