# OpenReview forum: "From Seeing to Doing: Bridging Reasoning and Decision for Robotic Manipulation"
_ICLR.cc/2026/Conference — ICLR 2026 Poster_

### Official Review · Reviewer_Kh3L · 2025-10-31

**Soundness:** 3
**Presentation:** 2
**Contribution:** 2
**Rating:** 6
**Confidence:** 4

**Summary:**

This paper introduces FSD, a vision-language model designed to bridge visual reasoning and robotic decision-making. FSD employs a Spatial Relationship-focused Chain-of-Thought (SrCoT) mechanism to generate intermediate visual representations (e.g., visual points and visual traces), enabling a structured transition from perception to action. It further integrates a hierarchical weak-to-strong data pipeline and a self-consistency alignment mechanism to enhance relation understanding and generation.

**Strengths:**

S1. Using visual prompts for zero-shot manipulation is a promising research direction.

S2. The benchmark proposed by the authors can help the robotics community evaluate the generation and reasoning capabilities of VLM models.

**Weaknesses:**

W1. Generating such a large number of visual prompts may affect inference efficiency and potentially obscure critical information. Moreover, I am not convinced that this model can achieve faster inference than end-to-end VLA models. The authors are encouraged to compare execution efficiency with Pi_0.

W2. How does the method address the closed-loop problem, for example, when the scene dynamically changes after generating visual prompts? Would redoing all prompt reasoning in such cases be too inefficient?

W3. Regarding the automated dataset construction, does it only build image-level planar relationships? Could the authors consider incorporating spatial relationships in the 3D Cartesian coordinate system?

W4. The end-to-end VLA scores reported on SimplerEnv (WidowX Robot) appear to be taken from SpatialVLA (Qu et al., 2025). It is recommended that the authors reproduce existing VLA models for fair comparison, include more recent VLAs, and report fine-tuned results as well.

W5. Although multiple visual and language prompts are generated, the accuracy does not reach the current SOTA. The authors should compare FSD with fine-tuned end-to-end VLAs, since such models are trained through imitation learning rather than a purely zero-shot paradigm.

**Questions:**

Q1. It is recommended that the authors explore more challenging tasks, such as articulated object manipulation or contact-rich manipulation scenarios.


I think this paper is currently borderline, but I’m willing to give it a borderline accept, provided that the authors address my concerns. If there is any misunderstanding on my review, I would appreciate it if the authors could clarify.

---

> ### Author Response · Authors · 2025-11-23
> **Response-Part 1**
>
> Dear Reviewer Kh3L:
>
> We thank the reviewer for the constructive feedback and for recognizing FSD's promising direction in using visual prompts for zero-shot manipulation and the value of our proposed benchmark. Below, we address the specific questions and concerns.
>
> ### **W1: Inference Efficiency & Visual Prompts**
>
> **[A. Misunderstanding regarding Visual Prompt Occlusion:]**
> We would like to clarify that the "visual prompts" (e.g., points, boxes) generated by FSD are **textual tokens (coordinates)** output by the VLM, representing the model's internal reasoning results. During the inference process, these markers are not overlaid on the input image fed to the VLM. Therefore, they strictly do not obscure or interfere with the VLM's perception of the original image information. We only visualize them on the image during debugging or for qualitative figure generation.
>
> **[B. Efficiency Comparison (vs. End-to-End VLA including pi0):]**
> Yes, your assessment is correct. We openly acknowledge that inference speed is not the primary advantage of FSD; this is an inherent characteristic shared by Affordance and Modular VLAs compared to highly optimized policies. The execution time comparison provided in our paper is intended to demonstrate that FSD executes significantly faster than Modular VLAs (e.g., MOKA) and maintains an **acceptable execution speed** compared to End-to-End methods, while offering **superior generalization capabilities**. A similar logic applies to the comparison with 𝜋0, particularly noting that action chunking in end-to-end models can reduce overall execution time.
>
> To better illustrate this point, we provide a detailed breakdown of module latencies measured on an A100 GPU:
>
> - **FSD (Ours - Affordance VLA):**
>     - **Total Perception Time:** ~4s.
>     - **Breakdown:** Model Inference (about 3-4s) + 2D-to-3D Mapping (< 50ms) + GraspNet (~100ms).
>     - **Mechanism:** FSD benefits from a **single-pass inference** mechanism, generating all necessary visual aids at the start of the task, which avoids cumulative latency during execution.
> - **MOKA (Modular VLA):**
>     - **Total Perception Time:** > 20s.
>     - **Breakdown:** Subtask Proposal (about 5s) + Filtering (about 4s) + Mark Annotation (about 1s) + Point Generation (about 10s).
>     - **Drawback:** MOKA suffers from severe cascading delays due to the sequential execution of multiple disjoint models.
> - **OpenVLA (End-to-End VLA):**
>     - **Per-Step Latency:** ~0.3s.
>     - **Drawback:** While a single step is fast, it must be executed continuously at high frequency (e.g., 10-20Hz). Over a typical task episode, the cumulative compute time is high, often resulting in slower, more hesitant motion compared to FSD's smooth trajectory execution.
>
> **Conclusion on Trade-off:** We acknowledge that for extremely long-horizon or highly dynamic tasks, the open-loop nature of FSD may require periodic re-planning, which would increase the time cost. However, our core contribution is demonstrating that FSD achieves a **superior balance between Generalization and Execution Efficiency**. FSD offers strong zero-shot capabilities with acceptable real-world latency. In our revised manuscript, we have added a discussion of this inference latency trade-off to the "Limitations" section (Appendix J), and we consider optimizing inference speed a priority for future work.
>
> ### **W2: Closed-Loop and Dynamic Scenes**
>
> **[A. Integration with Closed-Loop Architectures:]**
> We have proactively addressed the closed-loop requirement. In **Appendix K**, we demonstrate a hybrid integration where FSD is combined with a low-level Diffusion Policy. FSD provides the high-level visual trace as guidance, while the Diffusion Policy executes actions in a closed-loop manner. This combination achieved an 85.3% success rate on the Libero-Spatial benchmark, surpassing the pure closed-loop baseline.
>
> **[B.Regarding dynamically changing scenes]**
>
> Yes, in highly dynamic scenarios (e.g. need >100hz), pure open-loop execution is not applicable and multiple re-planning steps are required. **In such cases, most modular VLAs and Affordance VLAs will fail.** At this point, high-frequency closed-loop strategies are more suitable, while FSD can still provide task-level guidance to the closed-loop strategy through a small number of re-planning steps, thereby improving the success rate. Developing event-driven re-planning systems is also a promising research direction.

---

> ### Author Response · Authors · 2025-11-23
> **Response-Part 2**
>
> ### **W3: Automated Dataset Construction (2D vs 3D)**
>
> Our automated data pipeline already incorporates 3D spatial understanding, moving beyond simple planar relationships.
>
> - **3D Reconstruction:** As detailed in **Section 3.3** and **Appendix A**, we utilize some vision model to reconstruct the scene into a 3D point cloud.
> - **3D Geometric Reasoning:** The "Level 2: Spatial Relationship" data is calculated within this reconstructed 3D Cartesian coordinate system. For instance, we enforce a strict filter requiring at least a 20% relative depth gap between objects to confirm a spatial relationship.
>
> While the final input/output format uses 2D image coordinates for VLM compatibility, the underlying reasoning data is rigorously grounded in 3D geometry, ensuring the model learns physical spatial relations rather than just 2D pixel proximity.
>
> ### **W4: Simpler Results**
>
> Thank you for your suggestions. The results for Affordance VLAs and Modular VLAs were both reproduced by authors under the same settings. To ensure fairness, we re-examined the end-to-end VLA scores to confirm that all models were trained using the BridgeData dataset and then tested on SIMPLERENV. We reproduced OpenVLA and referring to recent articles[1], supplemented several classic baseline results. The complete new results have been updated in **Table 4 of the revision (page 9).**
>
> | Model | Type | Put Spoon on Towel | Put Carrot on Plate | Stack Green Block on Yellow Block | Put Eggplant in Yellow Basket | Avg |
> | --- | --- | --- | --- | --- | --- | --- |
> | Octo | End-to-end VLA | 41.7 | 8.2 | 0.0 | 56.7 | 26.7 |
> | $\pi 0$ | End-to-end VLA | 29.1 | 0.0 | 16.6 | 62.5 | 27.1 |
> | $\pi 0$-fast | End-to-end VLA | 29.1 | 21.9 | 10.8 | 66.6 | 48.3 |
> | OpenVLA | End-to-end VLA | 4.2 | 0.0 | 0.0 | 16.7 | 5.2 |
> | OpenVLA-OFT | End-to-end VLA | 34.2 | 30.0 | 30.0 | 72.5 | 41.8 |
> | MOKA | Modular VLA | 45.8 | 41.6 | 33.3 | 12.5 | 33.3 |
> | RoboPoint | Affordance VLA | 16.7 | 20.8 | 8.3 | 25.0 | 17.7 |
> | FSD | Affordance VLA | 41.6 | 50.0 | 33.3 | 37.5 | 40.6 |
>
>
> We have also updated our analysis. In this limited generalization testing scenario, the latest end-to-end VLAs perform better, but FSD still demonstrates competitive performance. We would also like to clarify that even when using the same embodiment and action space, end-to-end VLAs can collapse to a success rate of zero when faced with significant variations in background and instructions, whereas FSD is able to maintain its performance and exhibits excellent generalization. We believe that an important direction for future research is to investigate closed-loop policy VLAs explicitly guided by visual trajectories.
>
> [1] Chen et al. InternVLA-M1: A Spatially Guided Vision-Language-Action Framework for Generalist Robot Policy. Arxiv 2025.
>
> ### **W5: Compare FSD with Fine-tuned End-to-end VLAs**
>
> 1. Comparing FSD (a zero-shot generalist) against VLAs fine-tuned on specific tasks (imitation learning specialists) is unfair. Fine-tuned models simply "memorize" the training distribution, whereas FSD is designed to handle unseen scenarios. First, we believe that the SIMPLER experiments can provide partially referential performance results (Train: BridgeData, Test: SIMPLER WidowX). When the generalization requirements are not high, FSD can achieve competitive performance.
> 2. Furthermore, we conducted comparisons across two real-world tasks on the xArm6. OpenVLA represents the end-to-end approach, MOKA represents a modular VLM pipeline, and our FSD completes the entire visual trace generation process using a single model. We recorded the success rates. Notably, **OpenVLA required fine-tuning with 20 demonstrations, as its zero-shot success rate was 0%.**
>
> | **Task** | **OpenVLA (FT)** | **MOKA (ZS)** | **FSD (ZS)** |
> | --- | --- | --- | --- |
> | Pick up the sponge and place it outside the plate | **100%** | 40% | 80% |
> | Move the cucumber between the pot and the bowl | 40% | 60% | **80%** |
>
> **Results:** Notably, even after fine-tuning, OpenVLA performed poorly in Task 2 due to its inadequate generalization to position variations within the test distribution. In contrast, FSD demonstrated consistent performance under zero-shot conditions. Therefore, compared to "naïve" end-to-end methods, FSD’s paradigm of reasoning-driven SrCoT and intermediate visual aids offers advantages in both generalization and accuracy. We have added the experimental results to **Appendix M.3 (page31) in the revision of the paper**.

---

> > ### Author Response · Authors · 2025-11-23
> > **Response-Part 3**
> >
> > ### **Q1: Challenging Tasks**
> >
> > We have included challenging tasks in our evaluation and **added new demonstrations** **in the revision** to highlight the value of Visual Traces:
> >
> > - Our real-world evaluation includes **"Fold the towel"** (Figure 7, bottom row), which involves deformable object manipulation.
> > - New Challenging Tasks: **(Appendix N, page 32):** In the revision, we have added demo results for additional challenging task: **"Wiping the whiteboard"** (contact-rich) and **"Opening the drawer"** (articulated). These results further demonstrate FSD's potential to handle complex, contact-rich manipulation scenarios via precise visual trace generation.
> >
> > ---
> >
> > We hope these clarifications and our revisions have addressed your concerns. Thank you once again for your valuable time and expertise. We welcome any further questions or discussion.

---

> ### Comment · Reviewer_Kh3L · 2025-11-25
> **Response to the rebuttal**
>
> I believe the authors’ rebuttal is very careful and detailed. **However, I find that the proposed method still does not address the issues of slow inference time or the inability to handle dynamic object manipulation, which are concerns similar to those raised by other reviewers.** I recommend that the authors integrate object tracking or other efficient strategies to mitigate the inference-time overhead. In addition, the long inference time limits the practicality of the entire system. Therefore, I still consider this paper to be at a baseline rating, and I hope the authors can improve the overall system efficiency in the next version.

---

> > ### Author Response · Authors · 2025-11-26
> >
> > Dear Reviewer Kh3L,
> >
> > Thank you for your constructive feedback and for acknowledging the details in our rebuttal.
> >
> > We agree that addressing inference latency and handling dynamic manipulation are critical for the system's practicality. We truly appreciate your suggestion regarding the integration of object tracking and other efficient strategies. We will prioritize these improvements in the next version to enhance the overall system efficiency and robustness.
> >
> > Thank you again for your valuable time and recommendations.
> >
> > Best regards, The Authors

---

### Official Review · Reviewer_vmwj · 2025-10-31

**Soundness:** 3
**Presentation:** 3
**Contribution:** 2
**Rating:** 6
**Confidence:** 4

**Summary:**

The paper proposes FSD, a VLM-based framework designed to generate visual aids for robotic tasks. These aids—spatial affordance boxes/points and object-centric visual traces—are produced via a Spatial-relationship focused Chain-of-Thought (SrCoT) and a self-consistency alignment that binds coordinates to visual references. The model is trained using a hierarchical, weak-to-strong data construction pipeline, creating 300k SFT samples from multiple robot datasets to teach five distinct capability levels (grounding → spatial relations → spatial reasoning → affordances → traces). Experiments report strong results on five spatial-reasoning benchmarks, good object/free-space referencing, high accuracy on the new VABench for visual aids, and successful zero-shot manipulation in SimplerEnv and on an xArm across eight real-world tasks.

**Strengths:**

Clear Problem Decomposition: The paper's primary strength is its clean decomposition of the problem: reason first, then produces compact, embodiment-agnostic visual aids (boxes, points, traces). This abstraction (§3.1) translates well to zero-shot action by simplifying the downstream tasks of 2D→3D back-projection and motion planning.

Methodological Novelty: The Spatial-relationship focused Chain-of-Thought (SrCoT) is a novel and promising approach. By anchoring reasoning on spatial relationship graphs with strict object and coordinate bindings (<ref>, <box>, <point>), the method plausibly reduces hallucination and improves grounding—a common failure point for VLMs.

Systematic Data Pipeline: The development of an automatic, multi-source 300k SFT dataset is a significant engineering contribution. The five-level capability curriculum, which uses external geometry tools (Metric3D v2, WildCamera) and rule-based filters, is well-conceived. Ablations effectively demonstrate that both the SrCoT and the alignment components contribute to performance.

Comprehensive Evaluation: The paper is thoroughly evaluated. FSD achieves state-of-the-art or near-SOTA results across a wide range of benchmarks (CVBench, CRPE, SAT, BLINK, EmbSpatial) and shows strong performance on grounding tasks (RoboRefIt, Where2Place). The large margins on the newly proposed VABench, combined with a 72% success rate on 8 real tasks, provide compelling evidence for the method's efficacy.

**Weaknesses:**

Incomplete Compute and Latency Attribution: The paper makes claims about deployability but fails to provide a rigorous analysis of its computational cost. A single small latency table is insufficient. The review lacks a module-level breakdown of FLOPs, parameters, and wall-clock time for essential components (projector, LLM tokens, SrCoT decoding, visual-aid generation, 2D→3D lift, and planning). Without this, it is impossible to judge the method's cost-benefit trade-off against modular or end-to-end VLAs under matched hardware and batching conditions.

Unexplored 2D Trace Limitations: The core robotic primitive—a 2D trace lifted to 3D via depth and pinhole back-projection—is sensitive to error. The paper does not quantify how errors in camera calibration (focal length, principal point), depth estimation noise, or occlusions accumulate and affect the final 3D trajectory. While 2D limitations are acknowledged, the lack of robustness studies is a significant gap.

Unknown Data Quality and Bias: The 300k auto-constructed dataset is a potential source of unmeasured error. The paper does not report label noise rates, inter-annotator agreement for the 300 manual VABench items, or how errors from the data pipeline (eg, the 20% relative depth gap filter) propagate to the quality of the final SrCoT reasoning.

Limited Scope of Robotic Interaction: The evaluation, while broad, is shallow in its robotic scope. Most executions appear to be open-loop, with closed-loop feedback mentioned but not rigorously evaluated. There is no analysis of drift, re-planning, or recovery from grasp failures. Furthermore, the tasks do not explore multi-step long-horizon sequences, heavy clutter, moving distractors, or significant camera pose changes, all of which are critical for real-world autonomy.

Weak Justification for Design Choices: Several key hyperparameters are fixed without ablation (eg, T=8 trace points). The evaluation for visual aids on VABench relies on a mix of metrics (MAE/RMSE) and an unvalidated LLM-score proxy.

**Questions:**

Cost Attribution and Scaling Analysis: Provide a detailed table with parameters, FLOPs, and p50/p90/p99 latencies per module (ViT, projector, SrCoT, aid decoding, planning). This analysis should be benchmarked on the same hardware used for comparisons (eg, vs. OpenVLA/MOKA) and show throughput vs. image resolution and token budget.

Geometric Robustness Stress Test: Conduct simulation experiments that quantify the drop in task success rate against applied noise. This must include sensitivity analysis for: (i) camera intrinsic perturbations (±Δf, ±Δcx,cy), (ii) synthetic depth noise, and (iii) mis-calibration. Includes a qualitative gallery of failure cases where 2D traces misproject.

Data Quality Audit: Report spot-check precision/recall of the auto-generated grounding and relation labels against a manual ground truth. For VABench, report the inter-annotator agreement (eg, Cohen's Kappa or Fleiss' Kappa) to validate the 300 manual items. Release data filtering scripts for reproducibility.

Closed-Loop and Granularity Ablations:

Implement a simple closed-loop variant (eg, visual servoing or re-planning at the next waypoint) and compare its success and correction frequency against the open-loop baseline.

Justify the choice of T=8 waypoints by ablating performance (VABench and SimplerEnv) vs. T ∈ {4, 16} and an adaptive T.

Human-in-the-Loop Evaluation: Strengthen the VABench claims by complementing the LLM-score proxy with a human preference study (eg, ≥300 pairwise prompts) on visual trace plausibility, reporting 95% confidence intervals.

---

> ### Author Response · Authors · 2025-11-23
> **Response-Part 1**
>
> Dear Reviewer vmwj:
>
> We thank the reviewer for the comprehensive summary and for recognizing our work's clear problem decomposition, methodological novelty and systematic data pipeline. Below, we address the specific questions and concerns.
>
> ### W1 & Q1: Computational Cost and Latency Attribution
>
> We appreciate the request for a rigorous computational analysis. We would like to clarify that FSD utilizes a unified VLM architecture. The "modules" mentioned (e.g., projector, visual aid generation) represent different stages within a single autoregressive inference pass, rather than separate, disconnected components. Therefore, we evaluate the visual trace generation process as a holistic part of the model's inference.
>
> To provide a meaningful comparison, we calculated the inference latency for three typical VLA paradigms on A100 GPU hardware. We focus on latency rather than FLOPs, as robotic tasks prioritize end-to-end inference speed and real-world execution time (wall-clock time):
>
> * **FSD (Ours - Affordance VLA):** * **Total Perception Time: ~4s**.
>     * *Breakdown:* Model Inference (approximately 3-4s) + 2D-to-3D Mapping (< 50ms) + GraspNet (~100ms).
>     * *Mechanism:* FSD benefits from a single-pass inference mechanism, generating all necessary visual aids at the start of the task.
> * **MOKA (Modular VLA):** * **Total Perception Time: > 20s**.
>     * *Breakdown:* Subtask Proposal (approximately 5s) + Filtering (approximately 4s) + Mark Annotation (approximately 1s) + Point Generation (approximately 10s).
>     * *Drawback:* MOKA suffers from severe cascading delays due to the sequential execution of multiple disjoint models.
> * **OpenVLA (End-to-End VLA):** * **Per-Step Latency: ~0.3s**.
>     - *Drawback:* While a single step is fast, it must be executed continuously at high frequency (e.g., 10-20Hz). Over a typical task episode, the cumulative compute time is high, often resulting in slower, more hesitant motion compared to FSD's smooth trajectory execution.
>
> We acknowledge that for extremely long-horizon or highly dynamic tasks, the open-loop nature of FSD may require periodic re-planning (e.g., re-running inference every ~3-4 seconds), which increases the time cost. This is an inherent characteristic of affordance-based and modular VLAs compared to high-frequency policies. However, our core contribution is demonstrating that FSD achieves a superior balance between **Generalization** and **Execution Efficiency**. FSD offers strong zero-shot capabilities with acceptable real-world latency.
>
> In our revised manuscript, we have added a discussion of this inference latency trade-off to the "Limitations" section, and we consider optimizing inference speed a priority for future work.
>
> ### **W2 & Q2: Geometric Robustness and 2D Trace Limitations**
>
> We agree that naively back-projecting raw 2D points is sensitive to noise. However, our system is designed with specific mechanisms to enhance robustness:
>
> 1. **Optimization-based Interpolation:** We do not rely on the simple back-projection of noisy depth pixels. As detailed in **Appendix C**, we employ an optimization-based trajectory ****interpolation method, which effectively smooth out sensor noise and depth fluctuations.
> 2. **Real-World Stress Test:** We consider our extensive real-world experiments (**Figure 7 and 14**) to be a rigorous "stress test." Our setup utilizes a single Intel RealSense L515 camera, which inherently introduces challenges such as depth noise, occlusion during execution, and imperfect calibration. Despite these physical imperfections, FSD achieved a **72% success rate** across 8 diverse tasks. This empirical evidence demonstrates that the system is robust to the levels of geometric noise commonly found in practical deployments.

---

> > ### Author Response · Authors · 2025-11-23
> > **Response-Part 2**
> >
> > ### W3 & Q3: About Data Quality and Bias
> >
> > Ensuring high data quality was the priority of our pipeline construction.
> >
> > **Automated Data Quality:** We did not rely solely on automated scripts. For every large-scale embodied dataset processed, we implemented strict rule-based filtering followed by **human verification**. As stated in **Appendix A**, for the Visual Trace and Affordance datasets, we iteratively refined filtering criteria until manual inspection of subsets yielded an accuracy of **>95%** before generating the full dataset.
> >
> > **VABench Annotation Quality:** The 300 items in VABench were manually annotated by the authors using consistent annotation and evaluation rules to establish a reliable Ground Truth. Since these were expert-annotated rather than crowdsourced, they serve as a standard gold reference, making inter-annotator agreement metrics (like Kappa) less applicable than in subjective labeling tasks.
> > **Rationale for the 20% Depth Gap Filter:** **This filter is a deliberate design choice to ensure quality.** Given inherent error’s in depth estimation models, we utilize a >20% relative depth gap to ensure that two objects are distinctly separable in 3D space, yielding high-confidence relational data. The rigor of our process is evidenced by the yield rate: we processed over **1 million** data points but selected only **28k** high-quality samples for the final visual aid dataset.
> >
> > ### W4 & Q4: Closed-Loop Evaluation
> >
> > We respectfully clarify that our paper does include closed-loop evaluation. In **Appendix K**, we integrated FSD with a Diffusion Policy (DP) in a closed-loop setting, where FSD generates the visual trace to guide the DP. This system was evaluated on the LIBERO-spatial benchmark, achieving an 85.3% success rate, significantly outperforming the baseline DP (76.1%). This demonstrates FSD's versatility as both a fast open-loop solver and a high-level planner for robust closed-loop policies.
> >
> > We acknowledge that Appendix K presents a preliminary integration. We identify extending FSD to support dynamic re-planning (e.g., re-querying the model every  $N$ steps) to update visual aids based on real-time feedback as a key direction for future work to enhance robustness in dynamic environments.
> >
> > ### W5 & Q5: Some Design Choices
> >
> > **[A. Ablation on Trajectory Point Density:]**
> >
> > Thank you for your suggestion. We have added new experiments with different trajectory point densities. We fixed the model trained on stage 1 data as the base model, and then adjusted the training data for stage 2 using resampling to support the generation of 4, 8, and 16 points, respectively. We present the ablation results on VABench-V. Here are the results:
> >
> > | VABench-V | RMSE↓ | MAE↓ | GPT Score |
> > | --- | --- | --- | --- |
> > | FSD-4p | 102.1 | 82.6 | 5.0 |
> > | FSD-8p (default) | **78.3**   | **63.4** | **6.2** |
> > | FSD-16p | 85.2 | 67.1 | 5.7 |
> >
> > **Results:** The results indicate that using 4 points yields the lowest score due to the loss of prediction accuracy, while the prediction performance with 16 points slightly degrades and is inferior to that with 8 points. This may be attributed to the fact that a greater number of predicted points increases the difficulty of both instruction following and reasoning processes. We argue that more complex tasks require predicting additional points to characterize more intricate visual trajectories; however, for most tasks, 8 points can provide sufficient granularity to generate smooth trajectories, achieving a balance between performance and speed. Therefore, for the majority of manipulation tasks, we adopt  $*T=8*$  points as a design choice to balance granularity and token complexity.
> >
> > **All the new ablation experiments have been updated in Appendix M (page30) in the revision.**
> >
> > **[B. Validity of Evaluation Metrics]**
> >
> > Since VABench consists of manually annotated Ground Truth, the quantitative metrics we reported (RMSE and MAE) directly measure the deviation from human intent6. This provides a more objective assessment than subjective human preference studies.
> >
> > We employ GPT-Score merely as a supplement to handle the "one-to-many" nature of instruction following. It is guided by detailed rubrics (**Appendix D**) to assess feasibility.
> >
> > Ultimately, our real-world experiments (72% success rate) serve as the final functional verification, confirming the quality of our generation methods beyond visual inspection.
> >
> > ---
> >
> > We hope these clarifications and our revisions have addressed your concerns. Thank you once again for your valuable time and expertise. We welcome any further questions or discussion.

---

> > > ### Author Response · Authors · 2025-11-26
> > >
> > > Dear Reviewer vmwj,
> > >
> > > Thank you once again for your insightful comments and the time you have dedicated to our manuscript. Your feedback has been invaluable to us.
> > >
> > > We have submitted our rebuttal to address your concerns. As the discussion period is coming to a close, we are eager to receive your feedback on our response. If you have any follow-up questions, we would be happy to address them before the deadline.
> > >
> > > Best regards, The Authors

---

### Official Review · Reviewer_ezj7 · 2025-10-31

**Soundness:** 3
**Presentation:** 3
**Contribution:** 3
**Rating:** 6
**Confidence:** 4

**Summary:**

This paper proposes FSD, a vision-language model that provides intermediate visual representations for robotic manipulation, such as affordance regions and visual traces. It presents detailed information on how to construct the dataset and train the model. Extensive experiments demonstrate the strong performance of FSD across multiple benchmarks, as well as its zero-shot open-loop capabilities. The model, along with its detailed data generation pipeline, dataset, and training strategy, offers valuable contributions for general robotic manipulation.

**Strengths:**

1. This paper provides a large dataset with visual aids, which is essential for robotic manipulation tasks. The data generation pipeline is detailed and well-designed. The experiments are extensive, and the model demonstrates strong performance.
2. The paper proposes weak-to-strong training and a self-consistency alignment strategy, both of which are effective for model performance.
3. The evaluations on real-world manipulation tasks and simulations are thorough, and the performance is strong. Both open-loop and closed-loop settings are tested.

**Weaknesses:**

1. Inference time may become a bottleneck for real-world applications.
2. The open-loop inference pipeline relies on complex and specifically designed submodules such as GraspNet and depth estimation. The overall success rate is influenced by compounded errors from both FSD and the downstream modules.
3. This paper presents promising results on closed-loop robotic manipulation in Appendix K, where visual aids are generated by FSD at the beginning of the task. It would be beneficial to update these visual aids every few steps to make the entire pipeline more practical and effective in real-world scenarios.

**Questions:**

1. In Table 3b, the DINOv2 predictor is compared. How many parameters of this model? Is it the same model size as FSD?
3. In Table 3, the results show that self-consistency alignment helps improve performance. Do you have any failure cases without alignment and success cases with alignment, so that we can clearly see the effectiveness of the self-consistency alignment?
4. How long does it take to generate the dataset?
5. Table 5 shows the execution time. Is this time for the whole task execution? What is the FSD inference time? OpenVLA is the closed loop model, which call the model constantly. What is the time cost for each component in your pipeline?

---

> ### Author Response · Authors · 2025-11-23
> **Response-Part 1**
>
> Dear Reviewer ezj7:
>
> We sincerely thank you for the positive assessment of our work, particularly for recognizing the value of our dataset construction pipeline, the weak-to-strong training strategy, and the extensive experimental validation across both simulation and real-world settings. Below, we address the specific questions and concerns.
>
> ### **W1 & Q4: Inference Time and Latency Breakdown**
>
> **[A. Definition of Execution Time:]**
> We confirm that your understanding is correct. The times reported in **Table 5** represent the **total wall-clock time** for completing the task. To ensure a realistic and fair comparison across paradigms—End-to-End VLA (OpenVLA), Modular VLA (MOKA), and Affordance VLA (FSD). This metric includes model inference, external module processing (e.g., GraspNet), and the physical movement of the robot arm.
>
> **[B. Latency Comparison and Breakdown:]**
> Under this "total task time" metric, FSD outperforms both MOKA and OpenVLA. This advantage stems from FSD's single-pass inference mechanism. However, OpenVLA requires high-frequency, step-by-step inference throughout the episode. MOKA suffers from significant accumulated latency due to its cascading multi-model pipeline.
>
> To provide a more rigorous comparison, we provide a breakdown of the inference latency (excluding robot motion) for each paradigm on our A100 GPU hardware:
>
> - **FSD (Ours):** * **Total Perception Time: ~4s**.
>
> Model Inference: ~3-4s + 2D-to-3D Mapping: < 50ms + GraspNet: ~100ms
>
> - **MOKA:** * **Total Perception Time: ~20s**.
>
> Subtask Proposal: ~5s + Filtering: ~4s + Mark Annotation: ~1s + Point Generation: ~10s
>
> - **OpenVLA:** * **Per-Step Latency: ~0.3s**.
>
> ~0.3s *per step*.
>
> MOKA suffers from severe cascading delays, and its point generation steps often encounter failures that require retries, making it the most time-consuming. As for OpenVLA, while a single step is fast, it must be executed continuously. Therefore, the cumulative compute time is higher, and the policy execution becomes slower and more hesitant. **When open-loop policies are available, FSD is competitive with end-to-end VLA methods such as OpenVLA in terms of total execution time, and its execution efficiency is significantly superior to that of modular VLA methods (MOKA).**
>
> **[C. Future Look & Generalization vs. Efficiency Trade-off:]** We recognize that for extremely long-horizon or highly dynamic tasks, the open-loop nature of FSD may require periodic re-planning (re-running inference every ~3-4 seconds), which increases the time cost. Compared with high-frequency policies, this is an inherent characteristic of affordance-based and modular VLAs. In addition, end-to-end VLAs can employ action chunking strategies to further reduce inference time.
>
> However, our core contribution is demonstrating that FSD achieves a superior balance between **Generalization** and **Execution Efficiency**. **FSD offers strong zero-shot capabilities with acceptable real-world latency.** We believe that this provides a valuable alternative path for current VLA research. In our revised manuscript, we have added a discussion of the inference latency trade-off to the "Limitations" section, and we consider optimizing inference speed a priority for future work.
>
> ### **W2: Dependency on Submodules and Compound Errors**
>
> We agree that pretrained model carry a risk of compound errors. However, compared to traditional Modular VLAs that chain many fragmented models, FSD consolidates the core reasoning and perception into a single robust model, significantly reducing error propagation.
>
> As for comparison with End-to-End VLAs, FSD's primary advantage is its zero-shot task execution capability without fine-tuning. End-to-End VLAs often struggle to generalize to unseen objects or instructions without specific data collection and fine-tuning.
>
> Besides, Our extensive evaluation demonstrates that FSD's reasoning-driven visual aids provide highly accurate initial guidance, effectively mitigating downstream errors. FSD achieves a 72% success rate in real-world tasks, outperforming the strongest baseline by 30%. This empirically proves that the generated visual aids are robust enough to guide the submodules effectively.

---

> > ### Author Response · Authors · 2025-11-23
> > **Response-Part 2**
> >
> > ### **W3: Closed-loop Updates (Updating Visual Aids)**
> >
> > We fully agree with this insightful suggestion. In **Appendix K**, we demonstrated a preliminary integration of FSD with a closed-loop Diffusion Policy, yielding promising results. We believe that extending FSD to support dynamic replanning and utilizing visual aids as explicit guidance for downstream VLA are key directions for our future work. We are convinced that implementing a "re-query" mechanism every *N* steps will indeed enhance robustness in dynamic environments.
> >
> > ### **Q1: DINOv2 Predictor Size vs. FSD**
> >
> > The DINOv2 Predictor is significantly smaller than FSD (13B).
> >
> > - **Architecture:** As detailed in **Appendix E**, the DINOv2 predictor consists of a pre-trained DINOv2 visual encoder (ViT-L/14, approx. 300M parameters) and a T5-Base language encoder (approx. 220M parameters), followed by a Transformer decoder.
> > - **Paradigm Comparison:** We chose this baseline not to compare parameter efficiency, but to compare **paradigms**: *Direct Regression (DINOv2)* vs. *Reasoning-based Generation (FSD)*. The results in **Table 3b** show that even a specialized regression model built on a SOTA visual encoder cannot match FSD's performance. This highlights that spatial reasoning capabilities derived from the VLM backbone and SrCot are critical for generating accurate visual aids, rather than simple feature extraction.
> >
> > ### **Q2: Effectiveness of Self-Consistency Alignment**
> >
> > **Quantitative Impact:** First, the ablation study in **Table 3** confirms the importance of alignment. Removing it causes VABench-Point accuracy to drop from 61.82% to 55.92%, and increases Visual Trace RMSE from 78.26 to 80.48.
> >
> > **Qualitative Analysis:** Then, to illustrate the effect, we identified a specific failure case without alignment in the **Appendix M.4 and Fig.16 (page 32)** in the revision. We observed a distinct phenomenon of "Spatial Hallucination" in the model trained *w/o* alignment. The model's textual reasoning remains logically correct. For example, it accurately generates the text plan "Lift the blue block upwards to…". However, the generated coordinate points deviate significantly. This disconnect implies that without bidirectional alignment training, the model treats coordinates merely as abstract linguistic symbols rather than mapping them to actual visual features.
> >
> > In contrast, the proposed bidirectional training mechanism with self-consistency alignment enforces a strict correspondence between spatial coordinates and visual features, ensuring that the generated points accurately land on the intended semantic targets and mitigating coordinate drift.
> >
> > ### **Q3: Dataset Generation Time**
> >
> > Generating the high-quality dataset is a computationally intensive one-time process due to the complex multi-stage pipeline described in **Section 3.3** and **Appendix A**. It involves processing massive datasets (BridgeDataV2, RT-X, Droid) using heavy vision models (SAM, Metric3D) and querying GPT-4o for reasoning chains.
> >
> > Processing the complete embodied data to generate the 5-level hierarchy took approximately five days on 4090 GPUs. In addition, it also depends on number of parallels. We consider this a worthwhile investment as it provides the foundation for training a robust generalist model.
> >
> > **We commit to open-sourcing the complete training dataset to contribute to the academic community.**
> >
> > ---
> >
> > We hope these clarifications and our revisions have addressed your concerns. Thank you once again for your valuable time and expertise. We welcome any further questions or discussion.

---

> ### Author Response · Authors · 2025-11-26
>
> Dear Reviewer ezj7,
>
> Thank you once again for your insightful comments and the time you have dedicated to our manuscript. Your feedback has been invaluable to us.
>
> We have submitted our rebuttal to address your concerns. As the discussion period is coming to a close, we are eager to receive your feedback on our response. If you have any follow-up questions, we would be happy to address them before the deadline.
>
> Best regards, The Authors

---

> > ### Comment · Reviewer_ezj7 · 2025-11-27
> >
> > Thank you for the detailed response. All of my concerns have been answered with sound supporting materials. Although FSD is valuable, effective, and general, its open-loop nature and long inference time remain the main bottlenecks to applying it to many kinds of real-world applications. If there were more solid evidence that FSD can benefit real-time robotic systems, I would be open to raising my score. However, I have to keep my score as it is for now.

---

> > > ### Author Response · Authors · 2025-11-27
> > >
> > > Dear Reviewer ezj7,
> > >
> > > We sincerely appreciate your continued engagement and your acknowledgment that our supporting materials have effectively addressed your previous concerns. We fully understand and respect your remaining concern regarding the trade-off between FSD's "open-loop nature/inference latency" and "real-time application."
> > >
> > > While we acknowledge that FSD is not designed for high-frequency control, we respectfully argue that **inference speed is not the sole metric for "benefiting real-world systems."** FSD delivers critical benefits to robotic systems in four specific ways that purely real-time End-to-End models cannot currently match:
> > >
> > > ### **1. Superior Generalization in Unstructured Environments**
> > >
> > > In many real-world scenarios (e.g., home service, variable manufacturing), the primary bottleneck is **adaptability (success rate)**, not execution frequency.
> > >
> > > - **Limitations of Real-Time E2E:** As shown in our comparisons and widely observed in the community, real-time E2E models (e.g., OpenVLA or \pi_0) can be fragile. Performance often drops significantly (sometimes to 0%) when facing unseen object categories, novel instructions, or viewpoint shifts, necessitating expensive data collection and fine-tuning.
> > > - **FSD's Advantage:** As a robust "Generalist," FSD prioritizes zero-shot understanding over speed for the vast long-tail of tasks. By bridging the "seeing-to-doing" gap without fine-tuning, FSD enables robotic systems to function in novel scenarios where traditional real-time policies simply fail to act.
> > >
> > > ### **2. Enhancing Real-Time Closed-Loop Systems as a High-Level Planner**
> > >
> > > FSD is not mutually exclusive with real-time systems; instead, it significantly boosts their performance by acting as a high-level guide.
> > >
> > > - **Hybrid Architecture:** As demonstrated in **Appendix K** and **Figure 7 (page 30)**, FSD generates "Visual Traces" that serve as explicit, grounded conditions for a high-frequency Diffusion Policy.
> > >
> > > |  | Libero Spatial (10 Tasks Avg.)  |
> > > | --- | --- |
> > > | DP Baseline  | 76.1 ± 1.6  |
> > > | **DP + FSD** | **85.3 ± 1.2** |
> > >
> > > In the LIBERO-Spatial benchmark, this hybrid system (FSD guiding Diffusion Policy) achieved an **85.3%** success rate, significantly outperforming the baseline Diffusion Policy (**76.1%**). This proves that FSD directly benefits real-time systems by providing the spatial reasoning that low-level policies lack, effectively combining "Slow Thinking" with "Fast Acting".
> > >
> > > ### **3. A Zero-Shot "Data Factory" for Learning Real-Time Policies**
> > >
> > > Even for applications with extreme real-time constraints, FSD is immensely valuable as an automated data generator.
> > >
> > > The biggest hurdle for real-time E2E models is the scarcity of high-quality embodied data. **FSD can autonomously generate successful trajectories for diverse tasks without human teleoperation, drastically reducing costs.** These trajectories can then be used to train or fine-tune lightweight, real-time student policies. In this pipeline, FSD serves as a "Data Factory" that replaces manual labor, significantly lowering the barrier to building efficient real-time systems.
> > >
> > > ### **4. Interpretability & Safety**
> > >
> > > Unlike "black-box" E2E policies, FSD explicitly visualizes the intended path via the Visual Aids. In real-world deployment, this interpretability allows human operators or safety algorithms to verify the path *before* the robot moves. In collaborative environments, this pre-execution safety check is often as critical as execution speed.
> > >
> > > ### **Conclusion**
> > >
> > > We believe that FSD benefits real-world robotics by solving the **generalization** and **data scarcity** bottlenecks, which are as critical as inference latency. We remain committed to optimizing inference efficiency (e.g., via quantization or using smaller size model) and exploring the "Data Factory" paradigm in our future work.
> > >
> > > The field of Embodied AI is flourishing, with multiple promising paths emerging—each with its own strengths, weaknesses, and applicable scenarios. No single approach is currently dominant. We aim to make a meaningful exploration and contribute a specific step forward in solving the challenges of real-world deployment.
> > >
> > > Thank you again for reconsidering our contribution.

---

### Official Review · Reviewer_aF4E · 2025-11-05

**Soundness:** 3
**Presentation:** 3
**Contribution:** 2
**Rating:** 6
**Confidence:** 4

**Summary:**

This work presented FSD (From Seeing to Doing), a framework that connects visual reasoning with robotic manipulation using intermediate spatial representations. FSD tackles key challenges of data scarcity and heterogeneity through three core innovations: a Spatial Relationship-Focused Visual Chain-of-Thought for multi-step reasoning, a hierarchical weak-to-strong data pipeline, and a self-consistency mechanism that aligns spatial coordinates with visual signals. Experiments demonstrate FSD’s effectivenss across multiple spatial reasoning and visual aid benchmarks.

**Strengths:**

1. The motivation is clear, by identifying limitations of existing VLA approaches, i.e., data scarcity, and heterogeneity, and a principled solution is proposed through object-centric intermediate representations of visual aids.

2. Extensive experiments across 8 benchmarks for spatial reasoning, plus zero-shot manipulation in both simulation and real-world settings demonstrate thorough validation.

**Weaknesses:**

1. The approach generates 2D trajectories that must be lifted to 3D using depth cameras. This is a significant limitation acknowledged by authors but I still have some questions:
A.	The depth model needs to predict the depth of 2D trace, which is tricky in my understanding, as this kind of trace could be out of distribution for the depth model. How robust / accurate is the depth estimation?
B.	How does a 2D trace avoid the collilsion, as there is uncertainty when it’s unprojected to 3D?

2. Ablation study of parameters: No ablation on the number of hierarchy levels. No comparison of different VLM backbones. No study on how performance changes with trajectory point density
3. The model is built based on HVicuna-13B and CLIP-ViT-L, how about the performance of a naïve VLA model implementation, without visual aid? This can give us an insight about the improvement with visual aid.
4. It uses GPT-4o to generate reasoning chains for training, then evaluates using GPT-4o for scoring (VABench GPT-score), which potentially leads the model to overfit to GPT-4o's biases.

**Questions:**

Please refer to weaknesses.

---

> ### Author Response · Authors · 2025-11-23
> **Response-Part 1**
>
> Dear Reviewer aF4E:
>
> We sincerely thank you for the insightful feedback and for recognizing our work's clear motivation, novel solution and extensive experimental validation. We will address the noted questions below.
>
> ### **W1: Limitations of 2D-to-3D Lifting (Depth Accuracy & Collision)**
>
> **[A. Robustness of Depth Estimation:]**
>
> We would like to clarify a potential misunderstanding regarding "depth prediction." FSD does not employ a model to predict the depth of 2D trace points from RGB images. Instead, as described in **Section 4** and **Appendix C**, we obtain the initial depth values directly from the depth camera. This is a standard robotic vision pipeline involving pinhole camera back-projection, rather than a generative prediction task. Therefore, the issue of "out-of-distribution" prediction for depth does not apply here, as we are querying physical sensor readings.
>
> **[B. Collision Avoidance in 2D Traces:]**
>
> This is an excellent question. Our framework addresses collision avoidance at two distinct levels:
>
> 1. **Reasoning Level:** The FSD model explicitly incorporates collision avoidance into its reasoning process. During the thinking phase, the model infers a path that circumvents obstacles in the 2D plane. For instance, the reasoning step in **Figure 6** (Page 8) explicitly states, *"First, lift the carrot ... to clear obstacles"*. Consequently, this generates an arc-shaped trajectory rather than a direct path, implicitly avoiding obstacles.
> 2. **Execution Level:** During execution, we employ specific strategies to ensure safety. For Affordance Points, we utilize CuRobo, a collision-aware motion planner, to generate safe 3D paths. For Visual Traces, we apply a interpolation (detailed in **Appendix C**) to optimize intermediate depth values, smoothing the trajectory to prevent erratic movements ("flying points") and reduce collision risks.
>
> Finally, we acknowledge that as a fully zero-shot generalization approach, FSD cannot guarantee absolute collision avoidance in all edge cases; these failure modes are reflected in our reported success rates. We believe that predicting 3D trajectories combined with explicit 3D spatial reasoning is a valuable direction for future work to fundamentally resolve collision issues.
>
> ### **W2: Need More Ablation Studies**
>
> **[A. Ablation on Hierarchy Levels:]**
>
> We designed the 5-level capability hierarchy as a logical weak-to-strong curriculum, where Levels 1-3 (Spatial Understanding) serve as prerequisites for Levels 4-5 (Visual Aid Generation) . To validate the effectiveness of two-stage learning, we conducted a comparative experiment called "**FSD w/o Stage 1**". In this experiment, we skipped the foundational training (Levels 1-3) and instead trained the model directly on visual aid data (Levels 4-5). All other experimental settings remain unchanged.
>
> |  | RoboPoint | VABench-Point |
> | --- | --- | --- |
> | FSD | **45.8** | **61.8** |
> | FSD w/o Stage 1 | 33.2 | 42.3 |
>
> **Results:** The results show that the FSD significantly outperforms the FSD w/o Stage 1 variant. The first stage provides essential spatial understanding and coordinate anchoring abilities. These are essential for the reasoning process and the generation of visual aids.
>
> **[B. Ablation on Trajectory Point Density:]**
>
> Thank you for your suggestion. We have added new experiments with different trajectory point densities. We fixed the model trained on stage 1 data as the base model, and then adjusted the training data for stage 2 using resampling to support the generation of 4, 8, and 16 points, respectively. We present the ablation results on VABench-V. Here are the results:
>
> | VABench-V | RMSE↓ | MAE↓ | GPT Score |
> | --- | --- | --- | --- |
> | FSD-4p | 102.1 | 82.6 | 5.0 |
> | FSD-8p (default) | **78.3**   | **63.4** | **6.2** |
> | FSD-16p | 85.2 | 67.1 | 5.7 |
>
> **Results:** The results indicate that using 4 points yields the lowest score due to the loss of prediction accuracy, while the prediction performance with 16 points slightly degrades and is inferior to that with 8 points. This may be attributed to the fact that a greater number of predicted points increases the difficulty of both instruction following and reasoning processes. We argue that more complex tasks require predicting additional points to characterize more intricate visual trajectories; however, for most tasks, 8 points can provide sufficient granularity to generate smooth trajectories, achieving a balance between performance and speed. Therefore, for the majority of manipulation tasks, we adopt  $*T=8*$  points as a design choice to balance granularity and token complexity.

---

> > ### Author Response · Authors · 2025-11-23
> > **Response-Part 2**
> >
> > **[C. Comparison of VLM Backbones:]**
> >
> > Our core contributions are the SrCoT framework, data pipeline, and alignment mechanism, which are agnostic to the VLM backbone. We selected the LLaVA-1.5 architecture as it is a robust, widely-used open-source baseline.  So we can conduct a fair comparison using the same base LLM model and other open-source models (RoboPoint & SAT) in Table 1. We believe that a more powerful VLM backbone will yield better results, and scaling to stronger backbones is an important direction for our future work.
> >
> > **All the new ablation experiments have been updated in Appendix M.1 and M.2 (page30) in the revision.**
> >
> > ### **W3: Performance of Naïve VLA (Without Visual Aid)**
> >
> > We would like to clarify that comparisons with Naïve VLA (End-to-end VLAs) have been included. Naïve VLA corresponds to vision-language-action models that output actions in an end-to-end manner, such as OpenVLA and RT-2. Without fine-tuning for specific scenarios, end-to-end VLAs face a severe "seeing to doing gap" with a success rate of exactly 0%. While end-to-end VLAs offer superior control precision, their current generalization capability is extremely limited. In contrast, the core advantage of FSD lies in its zero-shot manipulation ability, it can successfully handle unseen scenarios, task instructions, and objects. As a more powerful affordance-based VLA, FSD enables effective integration of perception and decision in novel tasks.
> >
> > Furthermore, we conducted comparisons across two real-world tasks on the xArm6. OpenVLA represents the end-to-end approach, MOKA represents a modular VLM pipeline, and our FSD completes the entire visual trace generation process using a single model. We recorded the success rates. Notably, **OpenVLA required fine-tuning with 20 demonstrations, as its zero-shot success rate was 0%.**
> >
> > | **Task** | **OpenVLA (FT)** | **MOKA (ZS)** | **FSD (ZS)** |
> > | --- | --- | --- | --- |
> > | Pick up the sponge and place it outside the plate | 100% | 40% | 80% |
> > | Move the cucumber between the pot and the bowl | 40% | 60% | 80% |
> >
> > **Results:** Notably, even after fine-tuning, OpenVLA performed poorly in Task 2 due to its inadequate generalization to position variations within the test distribution. In contrast, FSD demonstrated consistent performance under zero-shot conditions. Therefore, compared to "naïve" end-to-end methods, FSD’s paradigm of reasoning-driven SrCoT and intermediate visual aids offers advantages in both generalization and accuracy. We have added the experimental results to **Appendix M.3 (page31) in the revision of the paper**.
> >
> > ### **W4: Potential Overfitting to GPT-4o**
> >
> > We would like to clarify that LLMs play entirely distinct roles in training and evaluation, and thus do not suffer from overfitting or bias.
> >
> > 1. **Distinct Tasks & Prompts**: ***Training Phase:*** GPT-4o is solely utilized to perform textual SrCoT reasoning. Critical ground truth data (e.g., coordinates of bounding boxes/points/trajectories) are strictly derived from our automated data pipeline, obtained via visual tools (such as SAM, Metric3D, etc.), rather than generated by GPT-4o. ***Evaluation Phase (GPT-Score)*:** GPT-4.1 (not GPT-4o) is employed as an evaluator to assess the visual rationality of generated trajectories overlaid on images based on detailed scoring criteria. The input modalities, prompts, and tasks are completely different from those in the training phase, thereby minimizing the risk of circular overfitting.
> > 2. **Validation via Objective Metrics**:The superiority of FSD has been verified by objective, non-LLM metrics. On the VABench-V dataset, FSD achieved the best RMSE and MAE scores **(Table 3b)**, which are used to measure geometric accuracy against manually annotated ground truth. The success rate in practical applications (72%) further validates the effectiveness of the proposed method, which is independent of any LLM bias. GPT-Score is only used as a supplementary metric to evaluate semantic feasibility.
> >
> > ---
> >
> > We hope these clarifications and our revisions have addressed your concerns. Thank you once again for your valuable time and expertise. We welcome any further questions or discussion.

---

> > > ### Comment · Reviewer_aF4E · 2025-11-24
> > >
> > > Thank you for the explanation. I will retain my original score, but I’ve increased my confidence. Best of luck.

---

> > > > ### Author Response · Authors · 2025-11-26
> > > >
> > > > Dear Reviewer aF4E,
> > > >
> > > > Thank you for your reply and for increasing your confidence in the assessment. We sincerely appreciate your valuable feedback and the time you dedicated to reviewing our work.
> > > >
> > > > Best regards, The Authors

---

### Meta-Review · Area_Chair_27M3 · 2026-01-07

**Summary:**

In first round review, reviewers raise major concerns about the efficiency/real world applicability and lack of closed-loop results.

The authors provide new results regarding the two points. Reviewers' concerns are solved. Thus, the final decision is acceptance.

**Reviewer Concerns:**

Reviewers' major concerns about the efficiency/real world applicability and lack of closed-loop results are solved and other technical detail issues are also answered.

**Reviewer Scores:**

All reviewers will keep their positive scores or even raise to higher positive scores.

---

### Decision · Program_Chairs · 2026-01-26

Accept (Poster)